# LoRA-FA: Efficient and Effective Low Rank Representation Fine-tuning

## Abstract

Fine-tuning large language models (LLMs) is crucial for improving their performance on downstream tasks, but full-parameter fine-tuning (Full-FT) is computationally expensive and memory-intensive. Parameter-efficient fine-tuning (PEFT) methods, such as Low-Rank Adaptation (LoRA), address this by optimizing only a small subset of parameters. However, LoRA may underperform Full-FT in certain scenarios due to the intrinsic limitations of its low-rank gradients. In this work, we reveal an asymmetric, collapsible structure in LoRA's update: the low-rank modification to $W$ can be reformulated as a single-layer linear regression, implying that one of the LoRA factors can be frozen without sacrificing expressivity. Leveraging this insight, we introduce LoRA-FA, which freezes the projection-down matrix $A$ and trains only the projection-up matrix $B$. We further close the gap to Full-FT by deriving closed-form gradient corrections that minimize the discrepancy between the induced low-rank gradient and the full gradient. Through extensive experiments on diverse benchmarks, including GLUE, GSM8K, MT-Bench, and HumanEval, we demonstrate that LoRA-FA consistently achieves comparable performance to existing PEFT methods and Full-FT. Experiments on system efficiency show that LoRA-FA significantly reduces activation memory consumption and computational workload in fine-tuning.

## 1 Introduction

Large language models (LLMs) have become a cornerstone of natural language processing (Brown et al., 2020; Touvron et al., 2023a; OpenAI, 2023; Anil et al., 2023), and fine-tuning pre-trained LLMs has been shown to be very effective to improve their performance on various downstream tasks (Liu et al., 2019; Wei et al., 2021) and to enable them to align with human intents (Ouyang et al., 2022; Bai et al., 2022). However, fine-tuning LLMs via full-parameter is prohibitively expensive, for example, fine-tuning a Llama3-70B (Dubey & et al., 2024) model with AdamW (Loshchilov & Hutter, 2017) requires more than 1 TB of GPU memory to store model parameters, gradients, and optimizer states (Rajbhandari et al., 2020). To reduce the memory cost of full-parameter fine-tuning, parameter-efficient fine-tuning (PEFT) methods have been proposed to update only a small fraction of parameters, such as adapter weights (Houlsby et al., 2019; Hu et al., 2022) and prompt weights (Li & Liang, 2021; Lester et al., 2021). Among these methods, low-rank adaptation (LoRA) (Hu et al., 2022) has been shown to achieve comparable performance to full-parameter fine-tuning (Full-FT), and has been widely used in many applications (Dettmers et al., 2023).

Specifically, LoRA adds a parallel low-rank adapter alongside the weight of a linear layer, as shown in Figure 1(b), where $W$ is the pre-trained weight, $A$ and $B$ are low-rank weights. Because LoRA freezes $W$ and only updates smaller matrices $A$ and $B$, its memory overhead for trainable parameters and corresponding gradients and optimizer states can be largely reduced, compared to Full-FT as shown in Figure 1(a), which can be viewed as updating $W$ and freezing $A$ and $B$.

Although LoRA demonstrates high efficiency, its fine-tuning performance remains inferior to Full-FT (Büyükakyüz, 2024; Meng et al., 2024; Wang et al., 2024c). Recent studies have sought to bridge this gap by either improving the initialization of the weights $A$ and $B$ or approximating the gradients of Full-FT. However, existing gradient approximation based studies generally overlook the asymmetric contributions of $A$ and $B$, leaving room for potentially separable optimization strategies. In this work, we identify the asymmetric nature of $A$ and $B$ in updating $W$: the low-rank update of

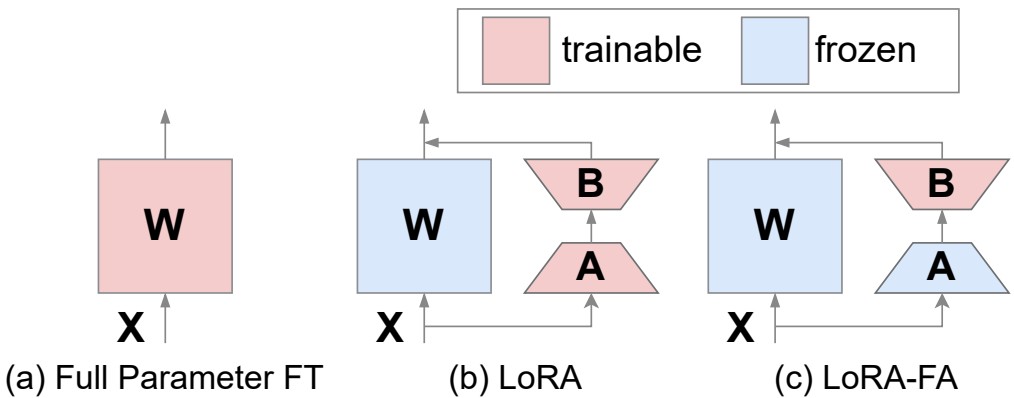

Figure 1: The illustration of (a) full-parameter fine-tuning (Full-FT), (b) LoRA, and (c) LoRA-FA.

$W$ in LoRA can be expressed as performing linear regression whose depth collapses to a single layer. Building on this insight, our objective is to approximate the gradient of Full-FT by updating only $A$ or $B$ (referred to as LoRA-FA and LoRA-FB, respectively). Formally, given $g^A$ and $g^B$ as the gradient of $A$ and $B$ respectively, this involves solving $\min_{g^A} \|\hat{g} - g\|_F^2$ and $\min_{g^B} \|\hat{g} - g\|_F^2$, where $\hat{g}$ denotes the gradient in updating $W$ when $A$ or $B$ is fixed. Through theoretical analysis, we demonstrate that both optimization problems yield independent closed-form solutions, illustrating that LoRA-FA and LoRA-FB can independently approximate the gradients of Full-FT with strategic gradient adjustments. Furthermore, we analyze the memory consumption of Full-FT and LoRA during training, revealing that LoRA-FA significantly reduces memory usage for activations compared to both LoRA and LoRA-FB. This positions LoRA-FA as an efficient and effective fine-tuning method.

Our main contributions are summarized as follows: (i) We first identify how $A$ and $B$ can be collapsed in updating $W$, showing that the low-rank update of $W$ in LoRA can be expressed as a single-layer linear regression. (ii) We propose LoRA-FA, an efficient and effective fine-tuning approach. By adjusting only a subset of gradients, LoRA-FA not only achieves competitive performance but also significantly reduces activation memory consumption and computational workload compared to other LoRA variants. (iii) We conduct extensive experiments, showing that LoRA-FA outperforms other PEFT methods in both fine-tuning performance and system efficiency across various models and datasets.

## 2 RELATED WORK

**Low-Rank Adaptation.** LoRA (Hu et al., 2022) is a widely adopted Parameter-Efficient Fine-tuning technique that incorporates low-rank adapters in parallel with the frozen layers of a model. This design yields substantial savings in optimizer state and gradient memory relative to full-parameter fine-tuning. Subsequent work extends LoRA along several axes. For instance, QLoRA (Dettmers et al., 2023) adapts LoRA for fine-tuning quantized models. Beyond quantization, a line of work modifies the learning dynamics of LoRA modules, e.g., altering learning rates for A versus B (Hayou et al., 2024), or sharing factors across layers (Kopiczko et al., 2024).

**Asymmetric LoRA.** Asymmetric LoRA approaches recognize that A and B contribute differently to the effective update and, consequently, admit different optimization and system properties. As shown by (Zhu et al., 2024), there is formal evidence that tuning A has limited importance when trying to match a desired output compared to tuning B. HydraLoRA (Tian et al., 2024) proposes using multiple parallel LoRA B modules with a single A module to improve performance. Based on this, LoRI (Zhang et al., 2025) freezes the A matrix and utilizes multiple B modules for multi-task adaptation. Notably, LoRI states that, in the limiting case of a single task and zero sparsity, it reduces to standard LoRA-FA without gradient approximation. In practice, asymmetric LoRA techniques are orthogonal to LoRA-FA, meaning they can be combined to further improve model adaptability.

**Gradient approximation for LoRA.** A complementary body of work aims to close the performance gap between LoRA and full fine-tuning by improving the low-rank gradient estimate rather than

modifying the module topology. LoRA-GA (Wang et al., 2024a) approximates the optimization trajectory of LoRA's first update step with that of full fine-tuning, and LoRA-Pro (Wang et al., 2024c) enhances the low-rank gradient in LoRA by approximating it with the full gradient. GoRA (He et al., 2025) leverages gradient information during training to dynamically assign optimal ranks and initialize low-rank adapter weights. LoRA Silver Bullet (Ponkshe et al., 2025) further approximates full fine-tuning within low-rank subspaces through a carefully designed initialization strategy.

Despite strong empirical adoption, existing LoRA variants are limited by the low-rank update subspace and by the need to retain forward activations for backpropagation through both A and B, which can dominate memory use and computational workload in long-context or large-batch regimes. LoRA-FA unifies the two lines above. It leverages the structural asymmetry of the A and B roles to freeze A, thereby collapsing depth and removing the need to store large intermediate activations. It also brings the gradient approximation perspective to bear by supplying a closed-form correction that aligns the induced low-rank update with the full gradient.

## 3 BACKGROUND AND MOTIVATION

Low-Rank Adaptation (LoRA) introduces a low-rank adapter alongside the weights of a linear layer, as described by the equation:

$$Y = XW_0 + \frac{\alpha}{r}XAB, \tag{1}$$

where $W_0 \in \mathbb{R}^{d_{in} \times d_{out}}$ represents the pre-trained weight matrix, $d_{in}$ is the input dimensionality, and $d_{out}$ is the output dimensionality. For simplicity, the bias term is omitted as it does not affect the analysis. Here, $X \in \mathbb{R}^{b \times s \times d_{in}}$ and $Y \in \mathbb{R}^{b \times s \times d_{out}}$ denote the input and output tensors, respectively, where $b$ is the batch size and $s$ is the sequence length. For the LoRA-specific components, $A \in \mathbb{R}^{d_{in} \times r}$ and $B \in \mathbb{R}^{r \times d_{out}}$ are the projection-down and projection-up weight matrices, respectively, with $r$ being the rank and $\alpha > 0$ a scaling hyperparameter. The factor $\frac{\alpha}{r}$ serves as the scaling coefficient of the product $AB$.

From Equation 1, the change in $W$ can be derived as follows:

$$dW = \frac{\alpha}{r}(dA\ B + A\ dB), \tag{2}$$

where $W = W_0 + \frac{\alpha}{r}AB$, $dA$ and $dB$ are given by their respective negative gradients. By the chain rule, the gradients of $A$ and $B$ in LoRA can be derived as:

$$g_{\text{LoRA}}^A = \frac{\partial L}{\partial W}\frac{\partial W}{\partial A} = \frac{\alpha}{r}gB^T,\ g_{\text{LoRA}}^B = \frac{\partial L}{\partial W}\frac{\partial W}{\partial B} = \frac{\alpha}{r}A^Tg \tag{3}$$

where $L$ and $g$ denote the loss function and the gradient of $W$, respectively. Recent studies Wang et al. (2024c); Meng et al. (2024) have shown that the rank of the gradient subspace for LoRA compared to full fine-tuning (Full-FT) (i.e., $2r \ll d$) is a key factor contributing to the performance gap between these approaches. In essence, LoRA leverages a low-rank gradient, $\hat{g}_{\text{LoRA}} = \frac{\alpha}{r}(g_{\text{LoRA}}^A B + Ag_{\text{LoRA}}^B)$, which combines the gradients of $A$ and $B$ to update $W$. To bridge the performance gap between LoRA and Full-FT, studies such as Wang et al. (2024a;c) have attempted to approximate the full gradient using the aforementioned low-rank gradient. Specifically, the optimization objective is to minimize $\|\hat{g} - g\|_F^2$, where $g$ represents the full gradient, and $\hat{g}$ denotes the low-rank gradient, which, in the case of LoRA, corresponds to $\hat{g}_{\text{LoRA}}$.

However, we have identified a critical collapsing behavior in the gradients of $A$ and $B$ during the training process. In LoRA optimization, where $\Delta W = AB$, this formulation intuitively implies that the updates to $W$ are restricted to the $r$-dimensional subspace $\mathcal{S}$ spanned by $A$ and $B$. Furthermore, if there exists another matrix pair spanning the same subspace $\mathcal{S}$, either $A$ or $B$ can be fixed, thereby reducing the trainable parameters to only a single component. To further elucidate this, we introduce Theorem 3.1, which shows that LoRA's update of $W$ can be expressed as a low-rank update achieved by a single trainable linear adapter.

**Theorem 3.1.** *Consider the optimization process in LoRA, where $\Delta W = AB$. Let $A_0$ denote the initial value of $A$, and suppose $A^* \in \mathbb{R}^{m \times r}$ and $B^* \in \mathbb{R}^{r \times n}$ are the optimal solutions for $A$ and $B$ in the LoRA optimization. Assume that both $A_0$ and $A^*$ are full-rank matrices. Then, the low-rank update of $W$ in LoRA can be expressed as a single-layer linear regression, formulated as:*

$$\Delta W = A^* B^* \approx A_0 B'^* \tag{4}$$

*where $B'^*$ has the same dimensions as $B$, i.e., $B'^* \in \mathbb{R}^{r \times n}$.*

*Proof.* See subsection A.1. $\qquad\square$

To summarize, since stacked linear becomes linear regression and the depth collapses to just one layer, we have proved that the expressiveness of LoRA is equivalent to that of LoRA with only one side adapter is trainable. Furthermore, in the next section, we prove that when $A$ or $B$ is frozen, such low-rank gradients can be strategically adjusted to approximate the gradients of Full-FT by solving an optimization problem. Moreover, this optimization problem has a closed-form optimal solution.

## 4 Bridging the Performance Gap

In this section, we first focus on bridging the performance gap between LoRA-FA (i.e., freezing $A$ and fine-tuning $B$) and Full-FT. Unlike standard LoRA, as described in Equation 3 and Equation 2, LoRA-FA employs a low-rank gradient, $\hat{g}_{\text{LoRA-FA}} = \frac{\alpha}{r} A g^B$, to update $W$. To mitigate the performance gap, we aim to strategically adjust $g^B$, the gradient of matrix $B$, to minimize the discrepancy between $\hat{g}_{\text{LoRA-FA}}$ and the full gradient $g$. This leads to the following optimization problem:

$$\min_{g^B} \|\hat{g}_{\text{LoRA-FA}} - g\|_F^2,$$
$$\text{s.t.} \quad \hat{g}_{\text{LoRA-FA}} = \frac{\alpha}{r} A g^B, \tag{5}$$
$$\mathrm{d}L \leq 0,$$

where $\| \cdot \|_F$ denotes the Frobenius norm, and $\mathrm{d}L$ represents the change in the loss function when updating with the gradient $g^B$. The goal is to minimize the gradient discrepancy while ensuring that the loss function is non-increasing. We demonstrate that this optimization problem admits an optimal closed-form solution, as shown in Theorem 4.1.

**Theorem 4.1.** *Let $A \in \mathbb{R}^{m \times r}$ be a fixed full-rank matrix, $g \in \mathbb{R}^{m \times n}$ denote the gradient of the loss with respect to $W$, and $g^B \in \mathbb{R}^{r \times n}$ represent the gradient with respect to $B$ in LoRA-FA. The objective function in Equation 5 is minimized by*

$$g^B = \left(\frac{r}{\alpha}\right)^2 (A^\top A)^{-1} g_{LoRA\text{-}FA}^B, \tag{6}$$

*where $\hat{g}_{LoRA\text{-}FA} = \frac{\alpha}{r} A g^B$. The solution also satisfies $\mathrm{d}L \leq 0$.*

*Proof.* See subsection A.2. $\qquad\square$

More importantly, the optimal closed-form solution reveals that the gradient adjustment in LoRA-FA depends solely on $A$ and $g_{\text{LoRA-FA}}^B$. This implies that the initialization of $B$ imposes no restrictions, and $g^B$ can be directly adjusted based on the original gradient $g_{\text{LoRA-FA}}^B$. We also extend this analysis to LoRA-FB (i.e., freezing $B$ and fine-tuning $A$), which exhibits similar properties. Please refer to subsection A.3 for more details.

## 5 LoRA-FA: LoRA by Fixing $A$

In this section, we present LoRA-FA, a novel fine-tuning method that is both efficient and effective.

**LoRA-FA can bridge the performance gap with Full-FT.** Building upon previous analysis, it has been established that, the low-rank update of $W$ in LoRA can be expressed as achieved by a single trainable linear adapter. (Theorem 3.1). Furthermore, it has been demonstrated that the gradient of $B$ or $A$ can be strategically modified to align the performance of LoRA-FA with that of Full-FT, achieving superior results compared to standard LoRA (Theorem 4.1). Next, we provide a detailed analysis of the system efficiency of LoRA-FA and LoRA-FB.

**LoRA-FA is more memory efficient.** In LoRA-FA, both the base weight $W$ and the adapter weight $A$ are frozen, requiring only the computation of the gradient of $B$. This results in the need to store only the much smaller intermediate activation $XA$ during the feed-forward pass, thereby eliminating the memory overhead of storing $X$ as required in standard LoRA. In contrast, LoRA-FB freezes the adapter weight $B$, necessitating the storage of the full activation of $X$ to compute the gradient of $A$. To analyze this more formally, assume $X \in \mathbb{R}^{b \times d}$, $W \in \mathbb{R}^{d \times d}$, $A \in \mathbb{R}^{d \times r}$, and $B \in \mathbb{R}^{r \times d}$. As shown in Figure 1, the projection-down weight $A$ maps the $d$-dimensional input $X$ to the $r$-dimensional intermediate activation $XA \in \mathbb{R}^{b \times r}$. Since $r \ll d$, storing the activation $XA$ in LoRA-FA is significantly more memory-efficient than storing the full activation $X$ in LoRA-FB. Consequently, the memory requirements for storing activations are greatly reduced in LoRA-FA. A comparative analysis of the memory complexity of low-rank modules is provided in Table 1.

**LoRA-FA can achieve greater computational efficiency through kernel-level optimization.** Consider the gradient computation of $B$ in Theorem 4.1. Since matrix $A$ remains fixed throughout training in LoRA-FA, the inverse of $A^T A$ can be precomputed once during model initialization and reused thereafter. Given that $A^T A$ has dimensions $r \times r$, and $r$ is typically very small, the memory required to store $A^T A$ is negligible, approximately 8 KB when $r = 64$. Moreover, revisiting Theorem 4.1, the gradient $g^B$ requires only the current gradient of $B$ for its computation, implying that the computational graph remains

Table 1: Memory complexity comparison among full fine-tune (FT), LoRA, LoRA-FA and LoRA-FB for a single linear layer with 16-bit mixed-precision training. # TPs is the number of trainable parameters. $d$, $r$, $b$, $s$ are hidden dimension, LoRA rank, batch size, and sequence length, respectively. We calculate the weight (W), gradient (G), optimizer (O), activation (A) in the unit of Bytes.

| Method | # TPs | W | G | O | A |
|--------|-------|---|---|---|---|
| Full-FT | $d^2$ | $2d^2$ | $2d^2$ | $4d^2$ | $2bsd$ |
| LoRA | $2dr$ | $2(d^2 + 2dr)$ | $4dr$ | $8dr$ | $2bsd + 2bsr$ |
| LoRA-FB | $dr$ | $2(d^2 + 2dr)$ | $2dr$ | $4dr$ | $2bsd$ |
| LoRA-FA | $dr$ | $2(d^2 + 2dr)$ | $2dr$ | $4dr$ | $2bsr$ |

unchanged. This structural property naturally facilitates the application of operator fusion techniques, thereby enabling more efficient computation within the training pipeline.[1]

# 6 EXPERIMENTS

We conduct extensive experiments to evaluate the effectiveness of LoRA-FA across a range of benchmarks, including GLUE (Wang et al., 2019), MT-Bench (Bai et al., 2024), GSM8K (Cobbe et al., 2021), and HumanEval (Chen & Jerry Tworek, 2021). First, we compare LoRA-FA with several other PEFT methods by fine-tuning RoBERTa-base/large on the GLUE benchmark. This provides a preliminary assessment of LoRA-FA's effectiveness in fine-tuning encoder-only models. Subsequently, we focus on supervised fine-tuning (SFT) of state-of-the-art LLMs in the MATH, CODE, and CHAT domains, comparing the performance of LoRA-FA against its competitors on benchmarks like GSM8K, HumanEval, and MT-Bench. Additionally, we examine the system efficiency of various methods during fine-tuning, demonstrating that LoRA-FA significantly reduces activation memory usage while maintaining or even slightly enhancing Model FLOPS Utilization (MFU), unlike other methods that exhibit a decline in MFU. The models, datasets, metrics, and baselines used in our experiments are detailed below, with further experimental settings provided in Appendix D due to space constraints.

**Models.** We fine-tune a diverse selection of LLMs, including encoder-only models such as RoBERTa-base/large (Liu et al., 2019) for natural language understanding (NLU) tasks, and decoder-only models such as Llama2-7B (Touvron et al., 2023b) and Llama3-8B (Dubey & et al., 2024) for natural language generation (NLG) tasks.

**Datasets.** In alignment with prior studies (Wang et al., 2024a;c), our experiments span a variety of datasets tailored to specific task types. For NLU tasks, we use the GLUE (Wang et al., 2019) benchmark. For SFT in domain-specific contexts, we utilize MetaMath (Yu et al., 2023) for the

---

[1]Please refer to Appendix B for detailed algorithm.

MATH domain, CodeFeedback (Zheng et al., 2025) for the CODE domain, and WizardLM (Xu et al., 2024) for the CHAT domain.

**Evaluation Metrics.** Following the evaluation protocols of prior works such as QLoRA (Dettmers et al., 2023), LLM-Adapters (Hu et al., 2023), and LoRA-Pro (Wang et al., 2024c), we primarily assess the zero-shot performance of fine-tuned LLMs across various benchmarks. For GSM8K, accuracy is reported as the evaluation metric. For MT-Bench, GPT-4 (OpenAI, 2023) is employed to score the quality of the model's responses, with the first-turn score reported as the metric. For HumanEval, we report the PASS@1 metric to evaluate code generation performance.

**Baselines.** We compare the performance of LoRA-FA against Full-FT, the standard LoRA, and several recent PEFT methods, including LoRA-QV, QLoRA (Dettmers et al., 2023), Vector-based Adaptation (VeRA) (Kopiczko et al., 2024), PiSSA (Meng et al., 2024), LoRA+ (Hayou et al., 2024), AdaLoRA (Zhang et al., 2023b), DoRA (Liu et al., 2024), LoRA-GA (Wang et al., 2024a), LoRA-Pro (Wang et al., 2024c). Please refer to Appendix D for detailed introduction of baselines.

## 6.1 PERFORMANCE ON GLUE BENCHMARKS

Table 2: Performance comparison on the GLUE benchmark. The batch sizes for fine-tuning RoBERTa-base and RoBERTa-large are 64 and 32, respectively. The LoRA rank is set to 8 by default, and the sequence length is 128 for both models. "Avg." denotes the average result across all tasks. The best and second-best results are marked in **bold** and underline, respectively. We report the average performance and its standard deviation over three independent runs for each task.

| Model | Method | SST2 | MRPC | QNLI | COLA | RTE | STSB | Avg. |
|---|---|---|---|---|---|---|---|---|
| | | | | GLUE | | | | |
| | Full-FT | 94.7±0.3 | 90.0±0.2 | 79.1±0.2 | 92.3±0.1 | 77.1±0.3 | 90.7±0.2 | 87.3 |
| | LoRA | 94.0±0.1 | 83.3±0.0 | 69.0±0.4 | 81.6±0.1 | 74.8±0.3 | 88.8±0.3 | 81.9 |
| | LoRA-QV | **95.0±0.2** | 89.6±0.3 | 70.2±0.1 | **93.3±0.1** | 76.8±0.4 | **91.4±0.4** | 86.1 |
| | QLoRA | 92.2±0.2 | 82.9±0.2 | 65.0±0.1 | 82.4±0.3 | 56.1±0.2 | 83.2±0.3 | 77.1 |
| RoBERTa-base | VeRA | 92.5±0.4 | 89.4±0.4 | 67.2±0.3 | 91.7±0.0 | 76.9±0.1 | 89.3±0.1 | 84.5 |
| | AdaLoRA | 93.5±0.0 | 82.3±0.3 | 70.1±0.3 | 92.0±0.2 | 73.1±0.2 | 87.1±0.1 | 83.0 |
| | LoRA+ | 93.9±0.4 | 85.2±0.0 | 75.1±0.2 | 93.1±0.4 | 77.0±0.1 | 87.1±0.4 | 85.2 |
| | PiSSA | 94.3±0.3 | 84.7±0.3 | 73.0±0.0 | 92.9±0.1 | 77.0±0.4 | 89.9±0.4 | 85.3 |
| | DoRA | 94.0±0.3 | 83.6±0.3 | 71.1±0.2 | 92.9±0.2 | 75.1±0.3 | 87.2±0.2 | 84.0 |
| | LoRA-GA | 94.3±0.0 | 87.8±0.2 | **80.0±0.1** | 93.1±0.4 | 77.0±0.4 | 88.1±0.3 | 86.7 |
| | LoRA-FA | 94.1±0.0 | **90.4±0.2** | 79.9±0.4 | 92.3±0.2 | **77.6±0.0** | 91.1±0.1 | **87.5** |
| | Method | SST2 | MRPC | QNLI | COLA | RTE | STSB | Avg. |
| | Full-FT | **96.2±0.0** | 90.1±0.0 | 80.0±0.0 | 94.3±0.2 | **86.0±0.0** | 92.1±0.0 | 89.8 |
| | LoRA | 95.2±0.1 | 89.3±0.3 | 72.0±0.2 | 94.5±0.3 | 82.4±0.4 | 92.0±0.2 | 87.6 |
| | LoRA-QV | **96.2±0.3** | 90.3±0.2 | 72.0±0.1 | **94.8±0.2** | 85.2±0.1 | **92.3±0.3** | 88.5 |
| RoBERTa-large | QLoRA | 94.1±0.0 | 87.0±0.2 | 69.0±0.1 | 90.5±0.3 | 71.1±0.3 | 89.9±0.2 | 83.6 |
| | VeRA | 96.0±0.3 | 90.8±0.4 | 70.0±0.1 | 94.4±0.1 | 85.9±0.1 | 91.6±0.0 | 88.1 |
| | AdaLoRA | 94.9±0.3 | 88.9±0.2 | 71.9±0.1 | 93.0±0.3 | 81.9±0.4 | 90.0±0.3 | 86.8 |
| | LoRA+ | 95.9±0.4 | 90.9±0.2 | 77.0±0.1 | 94.1±0.3 | 84.0±0.3 | 91.9±0.2 | 89.0 |
| | PiSSA | 96.0±0.2 | 89.9±0.4 | 73.9±0.2 | 94.1±0.1 | 85.0±0.0 | 89.9±0.0 | 88.1 |
| | DoRA | 94.0±0.2 | 89.0±0.3 | 71.9±0.0 | 93.0±0.1 | 83.0±0.2 | 91.0±0.1 | 87.0 |
| | LoRA-GA | 95.1±0.1 | 90.9±0.3 | **80.0±0.1** | 94.0±0.3 | 84.1±0.3 | 92.1±0.2 | 89.4 |
| | LoRA-FA | 96.1±0.4 | **91.2±0.0** | **80.9±0.4** | 94.4±0.3 | 85.5±0.2 | 92.0±0.2 | **90.0** |

Following a similar approach to the related work (Dettmers et al., 2023), we utilize the pre-trained RoBERTa-base model with 125 million parameters and RoBERTa-large model with 355 million parameters to evaluate fine-tuning performance on the GLUE benchmark. Drawing inspiration from (Mangrulkar et al., 2022), we first conduct a hyperparameter search on the MRPC task to determine the optimal settings, which are subsequently applied to other tasks. The results, summarized in Table 2, demonstrate that LoRA-FA achieves performance comparable to, and in some cases surpassing, Full-FT. Specifically, LoRA-FA achieves the best results on MRPC and RTE when fine-tuning RoBERTa-base, and on MRPC and QNLI when fine-tuning RoBERTa-large. Surprisingly,

LoRA-FA attains an average accuracy of 87.5% with RoBERTa-base and 90% with RoBERTa-large, both of which exceed the performance of Full-FT. Furthermore, LoRA-FA consistently and significantly outperforms standard LoRA across both models.

## 6.2 Performance on Domain-Specific Tasks

In this section, we evaluate the performance of LoRA-FA on LLMs, focusing on dialogue generation, mathematical reasoning, and code generation capabilities (i.e. CHAT, MATH, and CODE). Our experimental setup follows the configuration used in LoRA-GA (Wang et al., 2024a) and LoRA-Pro (Wang et al., 2024c).

Table 3: Performance comparison on the MT-Bench, HumanEval, and GSM8K benchmarks. The default rank is set to 64, and we use a maximum sequence length of 1024 with a global batch size of 8. The best and second-best results are marked in **bold** and underline, respectively. We report the average performance and its standard deviation over three independent runs for each task.

| Model | Llama2-7B | | | Llama3-8B | | |
|---|---|---|---|---|---|---|
| Method | MT-Bench | HumanEval | GSM8K | MT-Bench | HumanEval | GSM8K |
| Full-FT | 5.3±0.2 | **35.3±0.0** | **59.5±0.2** | **8.2±0.1** | **66.1±0.2** | **78.2±0.4** |
| LoRA | 5.6±0.0 | 14.8±0.2 | 42.9±0.4 | 7.4±0.1 | 62.2±0.2 | 71.3±0.0 |
| QLoRA | 4.9±0.1 | 12.1±0.5 | 42.4±0.4 | 6.9±0.1 | 59.8±0.1 | 68.9±0.7 |
| VeRA | 5.0±0.0 | 12.0±0.4 | 42.9±0.4 | 7.0±0.2 | 59.8±0.3 | 69.8±0.0 |
| AdaLoRA | 5.5±0.2 | 17.2±0.1 | 51.0±0.2 | 7.1±0.1 | 62.1±0.4 | 70.8±0.0 |
| LoRA+ | 5.7±0.1 | 17.8±0.0 | 51.5±0.2 | 7.4±0.1 | 63.0±0.2 | 71.3±0.3 |
| PiSSA | 5.3±0.2 | 16.2±0.1 | 45.6±0.3 | 7.4±0.1 | 63.8±0.1 | 71.3±0.2 |
| DoRA | 5.9±0.1 | 19.0±0.4 | 52.2±0.3 | 7.5±0.1 | 63.9±0.1 | 72.1±0.1 |
| LoRA-GA(r=64) | 5.9±0.2 | 19.2±0.4 | 54.5±0.3 | 7.5±0.0 | 63.9±0.2 | 72.1±0.4 |
| LoRA-GA(r=128) | **6.1±0.0** | 23.1±0.4 | 55.1±0.4 | 7.6±0.0 | 64.1±0.2 | 72.5±0.2 |
| LoRA-Pro(r=64) | 5.8±0.1 | 22.0±0.4 | 55.7±0.1 | 7.5±0.1 | 64.5±0.5 | 73.3±0.0 |
| LoRA-Pro(r=128) | 6.0±0.1 | 33.1±0.3 | 56.6±0.4 | 7.6±0.2 | 64.8±0.7 | 74.1±0.1 |
| LoRA-FA(r=64) | 5.7±0.0 | 28.1±0.4 | 57.0±0.2 | 7.5±0.0 | 64.5±0.3 | 75.3±0.5 |
| LoRA-FA(r=128) | **6.1±0.0** | 33.9±0.2 | 57.3±0.3 | 7.6±0.1 | 65.0±0.3 | 75.6±0.3 |

The results presented in Table 3 underscore the strong performance of LoRA-FA. Notably, LoRA-GA, LoRA-Pro, and LoRA-FA all achieve significant improvements over the original LoRA. For instance, LoRA-FA yields performance gains of 0.5 on MT-Bench, 19.1 on GSM8K, and 14.4 on HumanEval when fine-tuning LLaMA2-7B, and gains of 0.2 on MT-Bench, 2.8 on GSM8K, and 4.3 on HumanEval when fine-tuning LLaMA3-8B. Furthermore, LoRA-FA consistently matches the performance of LoRA-Pro. These findings validate the effectiveness of the proposed LoRA-FA approach. Additionally, the performance of LoRA-FA exhibits a consistent upward trend as the rank increases.

The experimental results also indicate that, in most cases, LoRA-FA achieves a level of approximate full-gradient capability comparable to that of LoRA-GA and LoRA-Pro. Since LoRA-FA trains only matrix $B$, whereas both LoRA-GA and LoRA-Pro train both $A$ and $B$, resulting in LoRA-FA halving the number of trainable parameters under the same rank, thereby doubling parameter efficiency. For instance, when fine-tuning Llama2-7B and evaluating on MT-Bench, LoRA-FA, with a rank of 64, slightly lags behind LoRA-Pro and LoRA-GA due to having only half the trainable parameters ($5.7_{\text{LoRA-FA, rank=64}}$ vs. $5.9_{\text{LoRA-GA, rank=64}}$ vs. $5.8_{\text{LoRA-Pro, rank=64}}$). However, when the rank is increased to 128, LoRA-FA matches the trainable parameter count of the other two methods and achieves superior performance ($6.1_{\text{LoRA-FA, rank=128}}$ vs. $5.9_{\text{LoRA-GA, rank=64}}$ vs. $5.8_{\text{LoRA-Pro, rank=64}}$).

In summary, the results demonstrate that LoRA-FA doubles parameter efficiency without compromising performance and can further enhance performance by doubling the rank, thereby achieving both efficiency and scalability.

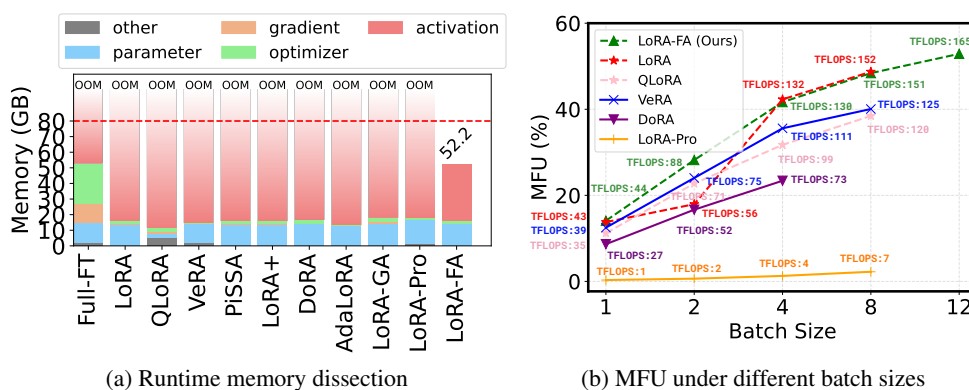

(a) Runtime memory dissection  (b) MFU under different batch sizes

Figure 2: (a): Dissecting the runtime memory overhead of different methods in fine-tuning Llama2-7B. We use a global batch size of 8, and a maximum sequence length of 1024. The dashed red line denotes the 80GB capacity of the NVIDIA A800 GPU. (b): Model FLOPs utilization (MFU) of different methods in fine-tuning Llama2-7B. To ensure the successful execution of other PEFT methods under most batch size settings, the sequence length is fixed at 512. The ranks for all methods in (a) and (b) are fixd to 64.

## 6.3 SYSTEM EFFICIENCY

### 6.3.1 ACTIVATION REDUCTION

To evaluate the system efficiency of LoRA-FA, we conducted an experiment comparing memory usage across a range of PEFT methods, including LoRA, QLoRA, VeRA, PiSSA, LoRA+, DoRA, AdaLoRA, LoRA-GA, LoRA-Pro, and our proposed LoRA-FA. We employed the benchmarking tool provided by (Zhang et al., 2023a) and performed the evaluation on a single NVIDIA A800 GPU. The experimental results are presented in Figure 2. The findings demonstrate that LoRA-FA consistently achieves substantial memory savings compared to other PEFT methods during fine-tuning. Specifically, LoRA-FA reduces memory usage by more than 27.8 GB (80 GB vs. 52.2 GB) relative to vanilla LoRA, making it the only method capable of running under the given memory constraints. In contrast, all other methods exceed the GPU memory limit, resulting in out-of-memory (OOM) errors.

Furthermore, the results highlight that activation memory constitutes the dominant component of memory consumption during training. Although QLoRA stores model parameters in 4-bit precision, its high activation memory cost also prevents it from running successfully. In particular, in addition to the standard forward activations required for backpropagation, QLoRA must retain both the quantization states and a copy of the quantized model to compute input gradients. As illustrated in Figure 2(a), it is important to note that the number of trainable parameters does not directly correlate with runtime memory efficiency.

### 6.3.2 COMPUTATIONAL EFFICIENCY

By eliminating the activation of $A$, LoRA-FA also removes both the feed-forward and backpropagation computations associated with the down-projection matrix $A$. Although LoRA-FA introduces additional computational overhead for computing the gradient of $B$, this overhead is mitigated through kernel-level optimization. To evaluate the efficiency of FLOPs utilization, we conduct a comparative experiment between LoRA-FA and other PEFT methods, measuring model FLOPs utilization (MFU) (Chowdhery & et al., 2022), as shown in Figure 2(b). The results demonstrate that LoRA-FA achieves MFU comparable to, or even higher than, that of LoRA, and consistently outperforms all other PEFT methods. Furthermore, due to its reduced memory footprint from eliminated activation storage, LoRA-FA supports larger batch sizes, scaling from 8 to 12, thereby enabling further speedups. Notably, our primary competitor, LoRA-Pro, exhibits MFU levels so low as to be nearly unusable, whereas LoRA-FA outperforms it in system efficiency by virtue of its lightweight computation.

## 6.4 DISCUSSIONS

Since matrix $A$ remains fixed in LoRA-FA, its initialization may influence the model's performance. To investigate this, we first examine the distributions of LoRA $A$ matrices corresponding to different base model modules before and after fine-tuning. As shown in Figure 3, when $A$ is initialized using a Gaussian distribution, all resulting $A^*$ matrices retain a Gaussian distribution post-training. In contrast, when initialized with a uniform distribution, the down-projection layer (i.e., the MLP's down-proj layer) maintains a strong Gaussian-like distribution, while other $A^*$ matrices exhibit a tendency to shift from a uniform to a Gaussian distribution during training. This observation suggests that, during training, $A$ tends to evolve toward a Gaussian distribution while preserving some characteristics of its initial distribution. To further assess the impact of initialization on LoRA-FA's performance, we investigate the following question: *if the initial $A_0$ follows a distribution similar to the final $A^*$, does this lead to improved performance?*

From the analysis above, we observe that regardless of the initialization method, the $A$ in the down-proj layer consistently exhibits a Gaussian distribution after training. Therefore, we test the hypothesis by initializing only the $A$ in the down-proj layer with a Gaussian distribution while keeping the other modules' $A$ initialized uniformly. The experimental results are reported in Table 4. They show that initializing the down-proj layer's $A$ to match the final distribution (i.e., Gaussian) yields improved performance (75.2 vs. 74.9). Furthermore, globally initializing all $A$ matrices with a Gaussian distribution also leads to better performance compared to global uniform initialization (75.6 vs. 74.9), supporting the importance of distribution alignment in LoRA-FA initialization.

Table 4: Ablation study on various initialization strategies for the LoRA-FA matrix $A$. "All Uniform" ("All Gaussian") indicates that every $A$ adapter is initialized with a uniform (Gaussian) distribution. "Uniform + D Gaussian" means only the $A$ adapter in the Down projection is initialized with a Gaussian distribution, while the rest are uniformly initialized.

| Model | Method | GSM8K |
|---|---|---|
| | All Uniform | 74.9 |
| Llama3-8B | Uniform + D Gaussian | 75.2 |
| | All Gaussian | 75.6 |

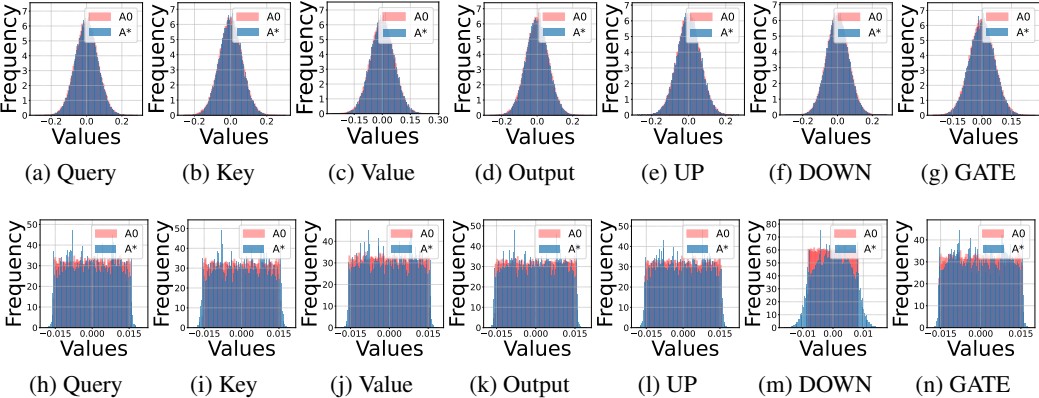

(a) Query    (b) Key    (c) Value    (d) Output    (e) UP    (f) DOWN    (g) GATE

(h) Query    (i) Key    (j) Value    (k) Output    (l) UP    (m) DOWN    (n) GATE

Figure 3: The distribution of the initial $A$ matrices ($A_0$) and the corresponding optimized $A^*$ across different layers during LoRA fine-tuning on LLaMA3-8B. Figures (a) - (g) represent cases where $A_0$ is initialized using a Gaussian distribution, while figures (h) - (n) correspond to cases where $A_0$ is initialized using a uniform distribution.

## 7 CONCLUSION

In this research, we present LoRA-FA, an efficient and effective PEFT approach. We first identify the collapsing behavior of $A$ and $B$ in updating $W$, showing that only a subset of the gradients needs to be approximated. Thus, by minimizing the discrepancy between the gradient of $B$ and full gradient $g$, while keeping the low-rank matrix $A$ frozen, LoRA-FA achieves performance on par with full fine-tuning while substantially reducing the memory and FLOPs..

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

## A PROOF OF THEORETICAL RESULTS

### A.1 PROOF OF THEOREM 2.1

**Theorem.** *Consider the optimization process in LoRA, where $\Delta W = AB$. Let $A_0$ denote the initial value of $A$, and suppose $A^* \in \mathbb{R}^{m \times r}$ and $B^* \in \mathbb{R}^{r \times n}$ are the optimal solutions for $A$ and $B$ in the LoRA optimization. Assume that both $A_0$ and $A^*$ are full-rank matrices. Then, the low-rank update of $W$ in LoRA can be expressed as a single-layer linear regression, formulated as:*

$$\Delta W = A^* B^* \approx A_0 B'^* \tag{7}$$

*where $B'^*$ has the same dimensions as $B$, i.e., $B'^* \in \mathbb{R}^{r \times n}$.*

*Proof.* In summary, the proof of Theorem 2.1 is divided into two parts. In the first part we derive the optimal solution, in the second part we give the Expected Value of the approximation under the distribution of $N(0, 1)$.

**Part I**

Given the optimal solutions $A^* \in \mathbb{R}^{m \times r}$ and $B^* \in \mathbb{R}^{r \times n}$, the optimal update to $W$ is given by:

$$\Delta W = A^* B^* \tag{8}$$

If there exists a matrix $C \in \mathbb{R}^{r \times r}$ such that $A^* = A_0 C$, substituting this into Equation 4 yields:

$$\Delta W = A^* B^* = A_0(C B^*) \tag{9}$$

Since the composition of linear transformations is equivalent to a single linear transformation, $C B^*$ can be reparameterized as $B'^*$. This demonstrates that the update represented in Equation 4 can be viewed as originating from a single-layer linear update.

Next, we derive the conditions under which such a $C$ exists:

1. $A^*$ and $A_0$ span the same subspace $\mathcal{S}$. If the two full-rank matrices $A^*$ and $A_0$ span the same subspace $\mathcal{S}$, there exists an invertible matrix $C \in \mathbb{R}^{r \times r}$ such that $A^* = A_0 C$.

2. $A^*$ and $A_0$ span different subspaces $\mathcal{S}' \neq \mathcal{S}$. If $A^*$ and $A_0$ span different subspaces $\mathcal{S}'$ and $\mathcal{S}$, we can approximate $C$ by solving the following optimization problem:

$$\min_C \|A^* - A_0 C\|_F^2 \tag{10}$$

where $\|\cdot\|_F$ denotes the Frobenius norm.

Let the objective function be:

$$f(C) = \|A^* - A_0 C\|_F^2 \tag{11}$$

Taking the gradient of $f$ with respect to $C$ and setting it to zero, we obtain:

$$\frac{\partial f}{\partial C} = -2 A_0^T (A^* - A_0 C) = 0 \tag{12}$$

$$A_0^T A_0 C = A_0^T A^* \tag{13}$$

Since $A_0^T A_0$ is invertible (as $A_0$ is full rank with rank $r$), we can solve for $C$:

$$C = (A_0^T A_0)^{-1} A_0^T A^* \tag{14}$$

This shows that $C$ has an optimal closed-form solution. Furthermore, since $A^*$ is the optimal solution for $A$ in the LoRA optimization, the corresponding $C$ is also optimal for updating $W$.

**Part II**

Next, we calculate the Expected Value of $\|A^* - A_0 C^*\|_F^2$ when $A^*$ and $A_0$ are under $N(0, 1)$.

First, recall that the residual can be expressed as:

$$A^* - A_0 C^* = A^* - A_0 (A_0^T A_0)^{-1} A_0^T A^*$$
$$= (I - P) A^*$$

where $P = A_0 (A_0^T A_0)^{-1} A_0^T$ is the projection matrix onto the column space of $A_0$, $I - P$ is the projection onto the orthogonal complement of the column space of $A_0$. Therefore, the residual $A^* - A_0 C^*$ is the projection of $A^*$ onto the orthogonal complement of $\mathrm{col}(A_0)$.

Since $A^*$ has entries from $N(0, 1)$, its columns $a_i^*$ are independent standard Gaussian vectors in $\mathbb{R}^m$. For each column $a_i^*$, the residual is:

$$r_i = (I - P) a_i^*$$

The squared Frobenius norm of the residual is:

$$\|A^* - A_0 C^*\|_F^2 = \sum_{i=1}^{r} \|r_i\|_2^2$$

Since the columns are identically distributed, it suffices to compute:

$$E\left[\|r_i\|_2^2\right], \quad \text{for any } i = 1, \ldots, r$$

Because $A^*$ and $A_0$ are independent and $A_0$ is fixed in each term when considering $a_i^*$, we condition on $A_0$. Let $\mathrm{col}(A_0)$ be the $r$-dimensional subspace of $\mathbb{R}^m$ spanned by the columns of $A_0$, $\mathrm{col}(A_0)^\perp$ be its orthogonal complement, which has dimension $m - r$. The projection $(I - P)$ projects any vector onto $\mathrm{col}(A_0)^\perp$.

Since $a_i^* \sim N(0, I_m)$, its variance in any direction is 1. The expected squared norm of its projection onto $\mathrm{col}(A_0)^\perp$ is:

$$E\left[\|r_i\|_2^2 \mid A_0\right] = \sum_{j=1}^{m-r} E\left[(v_j^T a_i^*)^2\right]$$

where $\{v_j\}$ is an orthonormal basis for $\mathrm{col}(A_0)^\perp$.

Since $v_j^T a_i^* \sim N(0, 1)$, we have:

$$E\left[(v_j^T a_i^*)^2\right] = 1$$

Thus:

$$E\left[\|r_i\|_2^2 \mid A_0\right] = m - r$$

Because this holds for each column $a_i^*$, the total expected squared norm is:

$$E\left[\|A^* - A_0 C^*\|_F^2 \mid A_0\right] = r \times (m - r)$$

Since $E\left[\|A^* - A_0 C^*\|_F^2 \mid A_0\right]$ does not depend on $A_0$ (the result is the same for any full-rank $A_0$), we have:

$$E\left[\|A^* - A_0 C^*\|_F^2\right] = r \times (m - r)$$

Therefore, the expected value is as above.

$$\square$$

**Remark 1.** The approximation $\Delta W = A^* B^* \approx A_0 B'^*$ should be interpreted as a structural approximation rather than an exact numerical equality. Although the expected residual $r(m - r)$ is not zero under Gaussian assumptions, this value represents the deviation in the ambient $m \times r$ space. In typical LoRA settings where $r \ll m$, the relative per-dimension discrepancy is small, and the projection $A_0 C^*$ preserves most of the subspace structure of $A^*$. Consequently, the low-rank update can still be effectively represented within the subspace of $A_0$, supporting the claim of a collapsible single-layer structure.

**Remark 2.** While the theoretical analysis often considers the case where A is initialized from a Gaussian distribution, this choice is primarily for analytical tractability and to align with common practice in neural network initialization. Importantly, our method and its theoretical guarantees are not fundamentally reliant on the exact distribution of A, as detailed in the following points: (i) Generality of Theoretical Results. In Theorem 4.1 and related proofs, the key requirement for the result to hold is that both the initial and optimal A matrices are full-rank and span the relevant subspace, not that they are strictly Gaussian. The Gaussian assumption is used in the expectation calculation to provide intuition about typical-case behavior, but the linear collapse and gradient adjustment results are distribution-agnostic as long as A is full-rank. (ii) Empirical Robustness to Initialization. Our ablation studies directly address this concern by evaluating LoRA-FA performance under different initializations of A (uniform, Gaussian, and mixed). The results demonstrate that LoRA-FA achieves strong and consistent performance regardless of whether A is initialized with a Gaussian or uniform distribution. Notably, the model tends to evolve A toward a Gaussian-like distribution during training, even when initialized otherwise (see Figure 3 and the related discussion).

A.2   PROOF OF THEOREM 3.1

**Theorem.** *Let $A \in \mathbb{R}^{m \times r}$ be a fixed full-rank matrix, $g \in \mathbb{R}^{m \times n}$ denote the gradient of the loss with respect to $W$, and $g^B \in \mathbb{R}^{r \times n}$ represent the gradient with respect to $B$ in LoRA-FA. The objective function*

$$\min_{g^B} \|\hat{g}_{LoRA\text{-}FA} - g\|_F^2 \tag{15}$$

*is minimized by*

$$g^B = \left(\frac{r}{\alpha}\right)^2 (A^\top A)^{-1} g^B_{LoRA\text{-}FA}, \tag{16}$$

*where $\hat{g}_{LoRA\text{-}FA} = \frac{\alpha}{r} A g^B$. The solution also satisfies $\mathrm{d}L \leq 0$.*

*Proof.* We aim to find $g^B$ that minimizes the Frobenius norm squared of the difference between $\frac{\alpha}{r} A g^B$ and $g$. Taking the gradient of $f$ with respect to $g^B$ and setting it to zero, we have

$$\frac{\partial f}{\partial g^B} = 2\frac{\alpha}{r} A^T \left(\frac{\alpha}{r} A g^B - g\right) = 0, \tag{17}$$

$$\implies \frac{\alpha}{r} A g^B = g. \tag{18}$$

Since $A$ is a full-rank matrix, its pseudo-inverse $A^\dagger$ is given by:

$$A^\dagger = (A^T A)^{-1} A^T \tag{19}$$

Using the pseudo-inverse, $g^B$ can be expressed as:

$$g^B = \frac{r}{\alpha} A^\dagger g = \frac{r}{\alpha} (A^T A)^{-1} A^T g \tag{20}$$

From Equation 3, the original gradient of $B$ is defined as:

$$g^B_{\text{LoRA-FA}} = \frac{\alpha}{r} A^T g \tag{21}$$

Substituting $g^B_{\text{LoRA-FA}}$ into the solution for $g^B$, we obtain:

$$g^B = (\frac{r}{\alpha})^2 (A^T A)^{-1} g^B_{\text{LoRA-FA}} \tag{22}$$

which provides the closed-form solution that minimizes $f$.

Next, we demonstrate that this solution also satisfies $\mathrm{d}L \leq 0$. We begin by showing that $\mathrm{d}L$ can be expressed as:

$$\mathrm{d}L = -\gamma\langle g_{\text{LoRA-FA}}^B, (\frac{r}{\alpha})^2(A^TA)^{-1}g_{\text{LoRA-FA}}^B\rangle_F \tag{23}$$

where $\gamma$ denotes the learning rate. To establish Equation 23, we first compute the differential change in the loss function, given by:

$$\mathrm{d}L = \langle\frac{\partial L}{\partial B}, \mathrm{d}B\rangle_F \tag{24}$$

Assuming $B$ is updated as $B = B - \gamma g^B$, then $\mathrm{d}B = -\gamma g^B$. Substituting $\frac{\partial L}{\partial B} = g_{\text{LoRA-FA}}^B$, we derive $\mathrm{d}L$ as:

$$\begin{aligned}
\mathrm{d}L &= -\gamma(\langle g_{\text{LoRA-FA}}^B, g^B\rangle_F) \\
&= -\gamma(g_{\text{LoRA-FA}}^B, (\frac{r}{\alpha})^2(A^TA)^{-1}g_{\text{LoRA-FA}}^B\rangle_F)
\end{aligned} \tag{25}$$

For any non-zero vector $x$, and given that $A$ is full-rank, we have:

$$\langle x, A^TAx\rangle = \langle Ax, Ax\rangle = \|Ax\|^2 > 0 \tag{26}$$

which implies that $A^\top A$ is positive definite. Consequently, $(A^\top A)^{-1}$ is also positive definite. Applying the Cholesky decomposition $(A^\top A)^{-1} = DD^\top$, we substitute this into $\mathrm{d}L$:

$$\begin{aligned}
\langle g_{\text{LoRA-FA}}^B, (\frac{r}{\alpha})^2(A^TA)^{-1}g_{\text{LoRA-FA}}^B\rangle_F &= (\frac{r}{\alpha})^2\langle g_{\text{LoRA-FA}}^B, DD^Tg_{\text{LoRA-FA}}^B\rangle_F \\
&= (\frac{r}{\alpha})^2\langle D^Tg_{\text{LoRA-FA}}^B, D^Tg_{\text{LoRA-FA}}^B\rangle_F \\
&= (\frac{r}{\alpha})^2\|D^Tg_{\text{LoRA-FA}}^B\|_F^2 \geq 0
\end{aligned} \tag{27}$$

Thus, $\mathrm{d}L \leq 0$, completing the proof. $\qquad\square$

### A.3 LoRA-FB and the proof of Theorem A.1

We extend our analysis to LoRA-FB (i.e., freezing $B$ and fine-tuning $A$), which exhibits similar properties. In LoRA-FB, the low-rank gradient is given as $\hat{g}_{\text{LoRA-FB}} = \frac{\alpha}{r}g^AB$. Our objective remains to minimize the discrepancy between $\hat{g}_{\text{LoRA-FB}}$ and the full gradient $g$. Accordingly, we formulate the following optimization problem:

$$\begin{aligned}
&\min_{g^A}\|\hat{g}_{\text{LoRA-FB}} - g\|_F^2, \\
&\text{s.t.}\quad \hat{g}_{\text{LoRA-FB}} = \frac{\alpha}{r}g^AB, \\
&\quad \mathrm{d}L \leq 0.
\end{aligned} \tag{28}$$

Following steps analogous to those used in proving Theorem 4.1, we provide the optimal closed-form solution to the optimization problem in Equation 28.

**Theorem A.1.** *Let $B \in \mathbb{R}^{r\times n}$ be a fixed full-rank matrix, $g \in \mathbb{R}^{m\times n}$ denote the gradient of the loss with respect to $W$, and $g^A \in \mathbb{R}^{m\times r}$ represent the gradient with respect to $A$ in LoRA-FB. The objective function*

$$\min_{g^A}\|\hat{g}_{\text{LoRA-FB}} - g\|_F^2 \tag{29}$$

*is minimized by*

$$g^A = \left(\frac{r}{\alpha}\right)^2 g_{\text{LoRA-FB}}^A(BB^\top)^{-1}, \tag{30}$$

*where $\hat{g}_{\text{LoRA-FA}} = \frac{\alpha}{r}g^AB$. The solution also satisfies $\mathrm{d}L \leq 0$.*

*Proof.* We aim to find $g^A$ that minimizes the Frobenius norm squared of the difference between $\frac{\alpha}{r}g^AB$ and $g$. Taking the gradient of $f$ with respect to $g^A$ and setting it to zero, we have

$$\frac{\partial f}{\partial g^A} = 2\frac{\alpha}{r}\left(\frac{\alpha}{r}g^AB - g\right)B^T = 0, \tag{31}$$

$$\implies \frac{\alpha}{r}g^AB = g. \tag{32}$$

Since $B$ is a full-rank matrix, its pseudo-inverse $B^\dagger$ is given by:

$$B^\dagger = B^T(BB^T)^{-1} \tag{33}$$

Using the pseudo-inverse, $g^A$ can be expressed as:

$$g^A = \frac{r}{\alpha}gB^\dagger = \frac{r}{\alpha}gB^T(BB^T)^{-1} \tag{34}$$

From Equation 3, the original gradient of $A$ is defined as:

$$g^A_{\text{LoRA-FB}} = \frac{\alpha}{r}gB^T \tag{35}$$

Substituting $g^A_{\text{LoRA-FB}}$ into the solution for $g^A$, we obtain:

$$g^A = (\frac{r}{\alpha})^2 g^A_{\text{LoRA-FB}}(BB^T)^{-1} \tag{36}$$

which provides the closed-form solution that minimizes $f$.

Next, we demonstrate that this solution also satisfies $\mathrm{d}L \leq 0$. We begin by showing that $\mathrm{d}L$ can be expressed as:

$$\mathrm{d}L = -\gamma\langle g^A_{\text{LoRA-FB}}, (\frac{r}{\alpha})^2 g^A_{\text{LoRA-FB}}(BB^T)^{-1}\rangle_F \tag{37}$$

where $\gamma$ denotes the learning rate. To establish Equation 37, we first compute the differential change in the loss function, given by:

$$\mathrm{d}L = \langle\frac{\partial L}{\partial A}, \mathrm{d}A\rangle_F \tag{38}$$

Assuming $A$ is updated as $A = A - \gamma g^A$, then $\mathrm{d}A = -\gamma g^A$. Substituting $\frac{\partial L}{\partial A} = g^A_{\text{LoRA-FB}}$, we derive $\mathrm{d}L$ as:

$$\begin{aligned}
\mathrm{d}L &= -\gamma(\langle g^A_{\text{LoRA-FB}}, g^A\rangle_F) \\
&= -\gamma(g^A_{\text{LoRA-FB}}, (\frac{r}{\alpha})^2 g^A_{\text{LoRA-FB}}(BB^T)^{-1}\rangle_F)
\end{aligned} \tag{39}$$

For any non-zero vector $x$, and given that $B$ is full-rank, we have:

$$\langle x, BB^T x\rangle = \langle B^T x, B^T x\rangle = \|B^T x\|^2 > 0 \tag{40}$$

which implies that $BB^T$ is positive definite. Consequently, $(BB^T)^{-1}$ is also positive definite. Applying the Cholesky decomposition $(BB^T)^{-1} = DD^\top$, we substitute this into $\mathrm{d}L$:

$$\begin{aligned}
\langle g^A_{\text{LoRA-FB}}, (\frac{r}{\alpha})^2 g^A_{\text{LoRA-FB}}(BB^T)^{-1}\rangle_F &= (\frac{r}{\alpha})^2\langle g^A_{\text{LoRA-FB}}, g^A_{\text{LoRA-FB}}DD^T\rangle_F \\
&= (\frac{r}{\alpha})^2\langle g^A_{\text{LoRA-FB}}D, g^A_{\text{LoRA-FB}}D\rangle_F \\
&= (\frac{r}{\alpha})^2\|g^A_{\text{LoRA-FB}}D\|^2_F \geq 0
\end{aligned} \tag{41}$$

Thus, $\mathrm{d}L \leq 0$, completing the proof. $\qquad\square$

**Remark.** Although $g$ (the gradient with respect to $W$) may not be directly accessible during training in LoRA-FA and LoRA-FB, AppendixA.2 and AppendixA.3 provide insight into how the gradient $g^B$ and $g^A$ can be adjusted using $g^B_{\text{LoRA-FA}}$, $A$ and $g^A_{\text{LoRA-FB}}$, $B$ to better approximate the full gradient update as in Full-FT.

# B  ALGORITHM OF LoRA-FA

---

**Algorithm 1:** AdamW with LoRA-FA

---

**Input:** LoRA scaling $\alpha$, rank $r$, learning rate $\eta$, AdamW coefficients $\beta_1, \beta_2$, weight decay $\lambda$
**Initialization:** $\{A_i\}_{i=1}^m$ with $A_i \sim \mathcal{N}(0, 1/r)$; $\{B_i\}_{i=1}^m = 0$; $\{m_i\}_{i=1}^m = 0$; $\{v_i\}_{i=1}^m = 0$

**for** $i = 1, \ldots, m$ **do**
    freeze $A_i$;
    $K_i \leftarrow (A_i A_i^\mathsf{T})^{-1}$ ;               // precompute inverse
**end**

**while** training **do**
    do forward and backward pass to obtain $\{\nabla_{B_i}\mathcal{L}\}_{i=1}^m$;
    **for** $i = 1, \ldots, m$ **do**
        $G_i \leftarrow \nabla_{B_i}\mathcal{L}$;
        $\tilde{G}_i \leftarrow \left(\frac{r}{\alpha}\right)^2 G_i K_i$ ;       // LoRA-FA gradient transform
        $m_i \leftarrow \beta_1 m_i + (1 - \beta_1)\tilde{G}_i$;
        $v_i \leftarrow \beta_2 v_i + (1 - \beta_2)\tilde{G}_i \odot \tilde{G}_i$;
        $\hat{m}_i \leftarrow \frac{m_i}{1 - \beta_1^t}$;
        $\hat{v}_i \leftarrow \frac{v_i}{1 - \beta_2^t}$;
        $B_i \leftarrow B_i - \eta \frac{\hat{m}_i}{\sqrt{\hat{v}_i} + \varepsilon}$;
        **if** $\lambda > 0$ **then**
            $B_i \leftarrow B_i - \eta \lambda B_i$ ;        // weight decay
        **end**
    **end**
**end**

---

# C  PROOFS RELATED TO MEMORY COMPLEXITY

In this section, we present a comprehensive memory complexity analysis in typical GPT-like fine-tuning. In general, the overall memory overhead consists of 4 categories: parameters, gradients, optimizer states, and activation.

**Denotations.**

- $L$ - number of layers
- $N$ - number of linear layers per layer
- $V$ - size of vocabulary
- $h$ - number of attention heads
- $b$ - batch size
- $s$ - sequence length
- $d$ - hidden dimension
- $r$ - LoRA rank
- $W$ - memory of parameter
- $G$ - memory of gradient
- $O$ - memory of optimizer state
- $A$ - memory of activation

## C.1 COMMON CONCEPTS

In this section, we introduce the common concepts of memory complexity on $W, G, O$ in Table 5. Specifically, $W$ depends on both the compute type and the number of parameter.

Table 5: Memory (in Bytes) of parameter, gradient, optimizer state, among Full-FT, LoRA, LoRA-FA, QLoRA, VeRA, in mix-precision fine-tuning. The model is loaded in 16-bit. #P denotes the number of parameter. #TP denotes the number of trainable parameter.

|  | Full-FT | LoRA | LoRA-FA | QLoRA | VeRA |
|---|---|---|---|---|---|
| #P | $12Ld^2 + dV$ | $12Ld^2 + dV + 16Ldr$ | $12Ld^2 + dV + 8Ldr$ | $12Ld^2 + dV + 16Ldr$ | $12Ld^2 + dV + 16dr + 8Lr$ |
| #TP | $12Ld^2 + dV$ | $16Ldr$ | $8Ldr$ | $16Ldr$ | $8Lr$ |
| $W$ | $24Ld^2 + 2dV$ | $24Ld^2 + 2dV + 32Ldr$ | $24Ld^2 + 2dV + 16Ldr$ | $6Ld^2 + \frac{dV}{2} + 16Ldr$ | $24Ld^2 + 2dV + 32dr + 16Lr$ |
| $G$ | $24Ld^2 + 2dV$ | $32Ldr$ | $16Ldr$ | $16Ldr$ | $8Lr$ |
| $O$ | $48Ld^2 + 4dV$ | $64Ldr$ | $32Ldr$ | $32Ldr$ | $16Lr$ |

## C.2 ACTIVATION MEMORY COMPLEXITY

In this section, we present the memory complexity per-layer in Bytes of Full-FT, in 3 data types, in Table 6.

Table 6: Activation memory complexity (in Bytes, per-layer) of Full-FT, in 3 data types. LN denotes the layernorm module.

| Data type | Attention | | | | | | | | MLP | | | | | Sum |
|---|---|---|---|---|---|---|---|---|---|---|---|---|---|---|
| | LN | QKV-i | QKV | softmax | dropout | matmul V | output | dropout | LN | up | gelu | down | dropout | |
| FP32 | 4bsd | 4bsd | 12bsd | 4bhss | bhss | 4bhss | 4bsd | bsd | 4bsd | 4bsd | 64bsd | 16bsd | bsd | 114bsd+9bhss |
| Pure-FP16 | 2bsd | 2bsd | 6bsd | 2bhss | bhss | 2bhss | 2bsd | bsd | 2bsd | 2bsd | 8bsd | 8bsd | bsd | 58bsd+5bhss |
| Autocast-BF16 | 4bsd | 2bsd | 6bsd | 4bhss | bhss | 2bhss | 2bsd | bsd | 4bsd | 2bsd | 5sd | 8bsd | bsd | 86bsd+7bhss |

## C.3 ACTIVATION MEMORY REDUCTION OF LoRA-FA COMPARED TO LoRA

In this section, we derive the activation memory savings achieved by LoRA-FA compared to LoRA when fine-tuning Llama2-7B.

In Llama2-7B, the Attention module consists of four linear layers: Query, Key, Value, and Output, each with dimensions $d \times d$. In the MLP module, there are three linear layers with dimensions $d \times 3d$, $3d \times d$, and $d \times d$, respectively. Consequently, LoRA-FA reduces the activation memory by a total of $18bsd$ per layer. When the number of layers is $L$, LoRA-FA reduces activation memory by at least $18bsdL$ bytes compared to LoRA.

For instance, when using a batch size of 8 and a sequence length of 1024, substituting $d = 4096$ and $L = 32$ for Llama2-7B, the theoretical savings of LoRA-FA over LoRA amount to at least 18GB of activation memory. In practice, due to PyTorch's memory reservation behavior or the retention of activations by other functions, the actual memory reduction achieved by LoRA-FA typically exceeds the theoretical estimate.

# D HYPERPARAMETERS AND EXPERIMENT SETTINGS

In this section, we present the baselines and hyperparameter used in the section 6 and Appendix E.

**Baselines**. We mainly compare the performance of LoRA-FA against Full-FT, the standard LoRA, and several recent PEFT methods, including the following:

- LoRA-QV: This method attaches the LoRA module only to the Query and Value layers, which aligns with the original setting in Hu et al. (2022).

- QLoRA (Dettmers et al., 2023): QLoRA is a quantized version of LoRA that applies 4-bit quantization to the base weights, significantly reducing memory overhead for parameter storage.

- Vector-based Adaptation (VeRA) (Kopiczko et al., 2024): VeRA uses a single pair of frozen low-rank matrices shared across all layers and updates a pair of vectors in each selected linear layer.

- PiSSA (Meng et al., 2024): PiSSA optimizes only the essential singular values and vectors while keeping the remaining components frozen.

- LoRA+ (Hayou et al., 2024): LoRA+ improves the learning rate for matrix $B$ in LoRA, allowing for more efficient feature learning based on theoretical insights.

- AdaLoRA (Zhang et al., 2023b): AdaLoRA dynamically adjusts the number of trainable parameters assigned to weight matrices and layers, optimizing parameter allocation.

- DoRA (Liu et al., 2024): DoRA employs weight decomposition to enhance performance, serving as a robust and effective baseline.

- LoRA-GA (Wang et al., 2024a): LoRA-GA approximates the optimization trajectory of the first step in LoRA to Full-FT, improving alignment with full fine-tuning.

- LoRA-Pro (Wang et al., 2024c): LoRA-Pro refines the low-rank gradient in LoRA by approximating it with the full gradient, aiming to close the performance gap with Full-FT.

Table 7: Hyperparameter configurations for finetuning different models on different datasets.

| Model | Dataset | Batch size | Rank | Sequence length | Learning rate |
|---|---|---|---|---|---|
| RoBERTa-base | GLUE | 64 | 8 | 128 | 4e-4 |
| RoBERTa-large | GLUE | 32 | 8 | 128 | 9e-5 |
| Llama2-7B | MetaMath | 32 | 64, 128 | 1024 | 3e-4 |
| Llama2-7B | CodeFeedback | 32 | 64, 128 | 1024 | 5e-5 |
| Llama2-7B | WizardLM | 32 | 64, 128 | 1024 | 5e-5 |
| Llama3-8B | MetaMath | 32 | 64, 128 | 1024 | 7e-5 |
| Llama3-8B | CodeFeedback | 32 | 64, 128 | 1024 | 5e-5 |
| Llama3-8B | WizardLM | 32 | 64, 128 | 1024 | 5e-5 |

Table 8: Hyperparameter configurations for fine-tuning large sequence length LLMs on consumer GPUs.

| Hyperparameter | RTX4090 | A800 (80GB) |
|---|---|---|
| # GPUs | 1 | |
| Batch size | 1 | |
| Seq. | 2048 | 4096 |
| LoRA layer | All linear | |
| LoRA-FA layer | All linear | |

Table 9: Hyperparameter configurations for effect evaluation of the number of LoRA-FA layers.

| # GPUs | Optimizer | Batch size | Rank | Seq. | LoRA-FA layer |
|---|---|---|---|---|---|
| 1 | AdamW | 1 | 64 | 1024 | All linear |

Table 10: Hyperparameter configurations for effect evaluation of the size of LoRA-FA rank.

| # GPUs | Optimizer | Batch size | Rank | Seq. | LoRA-FA layer |
|---|---|---|---|---|---|
| 1 | AdamW | 1 | 1, 2, 4, 8, 16, 32, 64, 128 | 1024 | All linear |

# E    MORE DISCUSSIONS

## E.1    ENLARGING SEQUENCE LENGTH IN MEMORY CONSTRAINT SCENARIO

In both pre-training and fine-tuning, it is clear that longer sequence lengths enhance performance (Dubey & et al., 2024; Abdin & et al., 2024; Wang et al., 2024b). However, the memory consumption for activations increases rapidly with the input size according to Table 1, making long-sequence training challenging. The efficient fine-tuning of LLMs can benefit significantly from memory optimization technologies. One such advancement is LoRA-FA, which, when combined with FlashAttention (Dao et al., 2022), demonstrates substantial improvements in memory efficiency. This combination is particularly advantageous for training LLMs with longer sequence lengths on GPUs with limited memory. Consumer-grade GPUs, such as the RTX 4090, have a memory capacity of 24GB, which is substantially less than 80GB on A800 or 40GB on A100 server-grade GPUs. This memory discrepancy poses a challenge for the efficient training of LLMs. To address this issue, we integrate FlashAttention with LoRA-FA to fine-tune the Llama2-7B model (Touvron et al., 2023b), aiming to evaluate the system's memory efficiency. For experiment settings, we set the sequence lengths to 2048 and 4096 for the RTX 4090 and A100 GPUs, respectively. We keep the rank fixed at 64, set the batch size to 1, and attach the LoRA layer to all linear layers. The results are detailed in Figure 4(a), which indicates that the combination of FlashAttention and LoRA-FA not only allows for the fine-tuning of LLMs with large sequence lengths on consumer-grade GPUs with 24GB memory, but also enables the extension of sequence length up to 4096 on GPUs with 40GB memory. This is a significant development, as LoRA with FlashAttention is unable to fine-tune such models due to the memory limit on the RTX 4090 GPU.

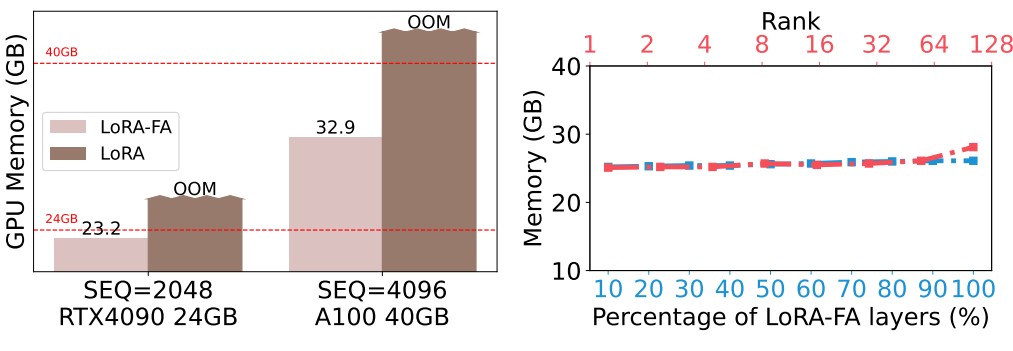

(a) Enlarge sequence length    (b) Rank, Percentage of LoRA-FA layers to memory

Figure 4: (a):Comparison of fine-tuning capability with large sequence length among LoRA-FA and LoRA, on RTX4090 (24GB) and A100 (40GB). In this experiment, we enable FlashAttention for both approaches. The rank is set to 64, batch size is set to 1, and LoRA is attached to all linear layers. The dashed red line denotes the 40GB and 24GB capacity of the NVIDIA A100 GPU and RTX4090 GPU respectively. (b):GPU memory footprint (GB) under different rank sizes and different number of LoRA-FA layer attached. We set the batch size to 1, and the sequence length to 1024. This result is from a single A800 80GB.

## E.2    MEMORY VS. LORA LAYERS AND RANK

**Effects of the number of LoRA-FA layers.** To understand how the integration of LoRA-FA layers impacts GPU memory usage, we conduct an experiment on the Llama2-7B model. Specifically, we investigate how varying the number (or percentage) of LoRA-FA layers affects the peak memory consumption during training. We set the batch size to 1, sequence length to 1024, and the rank to 64. Memory consumption with a changed number of adapters is shown in Figure 4(b), which shows that increasing the number of LoRA-FA layers does not lead to a rise in peak memory usage.

This outcome is significant because it indicates that LoRA-FA's memory footprint is not sensitive to the number of adapter layers implemented within the model. Consequently, it is feasible to attach LoRA-FA adapters to all linear layers within the model without worrying about escalating memory

requirements, while it aligns the accuracy with the Full-FT method. This capability is crucial for fine-tuning since it allows for maximal flexibility and the potential for enhanced model performance without compromising on memory efficiency.

**Effects of the rank in LoRA layers.** To elucidate how the rank in LoRA-FA affects GPU memory usage during fine-tuning, we conduct an analysis by varying the rank size while keeping other variables constant in fine-tuning Llama2-7B. The batch size is set to 1, the sequence length to 1024, and sweeping the rank size from 1 to 128. The impact of rank size on the memory footprint is captured in Figure 4(b). The results indicate that the GPU memory footprint is not significantly affected by changes in the rank size for LoRA-FA. This observation is noteworthy because it demonstrates that practitioners can adjust the rank size within LoRA-FA from the minimal value of 1 to as high as 128 without encountering out-of-memory (OOM). This flexibility allows for the fine-tuning of the model to be optimized for performance without typical memory constraints. Moreover, since LoRA-FA exhibits limited sensitivity to the rank in terms of memory consumption, it is always feasible to use higher LoRA ranks, which may potentially yield improved performance. In Table 3 of our paper, we specifically compare the performance at ranks 64 and 128, and the results demonstrate that performance remains consistently strong at rank 128. Therefore, for users employing LoRA-FA, we recommend directly adopting a rank of 128.

### E.3 INITIALIZATION OF A

In Table 4, we explored the initialization of A to demonstrate the relationship between $A_0$ and $A^*$. Many recent methods focus on the informative initialization of A to enhance the performance. Here, we take PiSSA as an example and delve into how LoRA-FA behaves if it adopts a PiSSA-style initialization. To address this, we adopted a PiSSA-style initialization for $A$ in LoRA-FA and evaluated the performance of Llama3-8B on the MATH domain, using a rank of 128. As shown in Table 11, PiSSA-style initialization is not well-suited for LoRA-FA. The underlying reason is straightforward: after the PiSSA decomposition, $A$ and $B$ are designed to be adapted jointly; thus, freezing $A$ is inherently incompatible with this initialization scheme. In the following part, we will provide a detailed theoretical derivation to further elucidate this phenomenon.

Table 11: Effect of initialization on LoRA-FA. PiSSA-style initialization underperforms standard Gaussian initialization on GSM8K with Llama3-8B (rank 128).

| Method | GSM8K Accuracy |
|---|---|
| LoRA-FA w/ Gaussian | 75.6 |
| LoRA-FA w/ PiSSA style | 70.2 |

Recall that in LoRA-FA one takes a single step

$$\Delta W = -\tfrac{\alpha}{r} A(A^T A)^{-1} A^T g = -\tfrac{\alpha}{r} P_A g,$$

where

$$P_A = A(A^T A)^{-1} A^T \in \mathbb{R}^{m \times m}$$

is the orthogonal projector onto $\mathrm{col}(A)$. Define the update energy by

$$E_A(g) := \|\Delta W\|_F^2 = \left(\tfrac{\alpha}{r}\right)^2 \|P_A g\|_F^2.$$

We compare two initializations of $A$. In truncated-SVD init, since

$$W \approx U_r \Sigma_r V_r^T, U_r \in \mathbb{R}^{m \times r}, \Sigma_r \in \mathbb{R}^{r \times r}, V_r \in \mathbb{R}^{n \times r},$$

then

$$A_{\mathrm{SVD}} = U_r \Sigma_r^{1/2}, B_{\mathrm{SVD}} = \Sigma_r^{1/2} V_r^T,$$

so that $A_{\mathrm{SVD}} B_{\mathrm{SVD}} \approx W$.

In Gaussian-random init, draw $A_{\mathrm{rand}}$ with i.i.d. $\mathcal{N}(0,1)$ entries. Equivalently, $\mathrm{col}(A_{\mathrm{rand}})$ is a uniformly random $r$-plane in $\mathbb{R}^m$.

We can derive such theorems:

**(i) SVD can have a "blind spot".** If $A_{\text{SVD}} = U_r \Sigma_r^{1/2}$, then $\text{col}(A_{\text{SVD}}) = \text{span}(U_r)$. Hence one may choose a nonzero gradient $g$ with every column in $\left(\text{span}(U_r)\right)^{\perp}$. For that $g$,

$$P_{A_{\text{SVD}}} g = 0 \implies E_{A_{\text{SVD}}}(g) = 0.$$

Proof: By construction $\text{col}(A_{\text{SVD}}) = \text{col}(U_r)$. If one picks any nonzero $g$ whose every column lies in the orthogonal complement of $\text{span}(U_r)$, then $P_{A_{\text{SVD}}} g = 0$ and so $E_{A_{\text{SVD}}}(g) = 0$.

**(II) Random subspace captures in expectation an $r/m$ fraction of any $g$.** Let $g = [g_1 \cdots g_n]$ with each $g_j \in \mathbb{R}^m$. For a uniformly random $r$-plane $S \subset \mathbb{R}^m$, a classical fact is

$$\mathbb{E}_S \left\| \text{Proj}_S x \right\|_2^2 = \frac{r}{m} \|x\|_2^2, \forall x \in \mathbb{R}^m.$$

Applying this column-wise gives

$$\mathbb{E}_{A_{\text{rand}}} \left\| P_{A_{\text{rand}}} g \right\|_F^2 = \sum_{j=1}^{n} \mathbb{E} \| P_{A_{\text{rand}}} g_j \|_2^2 = \frac{r}{m} \sum_{j=1}^{n} \|g_j\|_2^2 = \frac{r}{m} \|g\|_F^2.$$

Hence

$$\mathbb{E}_{A_{\text{rand}}} \left[ E_{A_{\text{rand}}}(g) \right] = \left( \frac{\alpha}{r} \right)^2 \frac{r}{m} \|g\|_F^2.$$

Proof: Writing $g$ column-wise and using the Grassmann-integration lemma,

$$\mathbb{E}_{A_{\text{rand}}} \left\| P_{A_{\text{rand}}} g \right\|_F^2 = \sum_{j=1}^{n} \mathbb{E} \left\| \text{Proj}_S g_j \right\|_2^2 = \frac{r}{m} \sum_{j=1}^{n} \|g_j\|_2^2 = \frac{r}{m} \|g\|_F^2,$$

and the factor $(\alpha/r)^2$ carries through to the definition of $E$.

In conclusion, the truncated-SVD initialization aligns $A$ with the top-$r$ eigendirections of $W$, which can be a good heuristic if one expects task gradients to lie in that subspace, but it admits worst-case gradients $g$ that it misses entirely ($E = 0$). However, the Gaussian-random initialization never has a nontrivial blind spot. On any fixed gradient $g$, it captures in expectation an $\frac{r}{m}$-fraction of $\|g\|^2$. Thus, for diverse downstream tasks (where $g$ may be arbitrary), random-init is strictly safer: it guarantees nonzero update energy on every gradient (with probability 1), and on average preserves a fixed fraction of gradient energy. In practice, to avoid worst-case blind spots and to ensure steady coverage of all gradient directions, Gaussian-random initialization is the better default choice for LoRA-FA.

### E.4 PERFORMANCE ON RECENT LARGER BASE MODELS

We conduct additional experiments on Qwen3-8B (Yang & Anfeng Li, 2025) and DeepSeek-V2-Lite-Chat (16B MoE) (DeepSeek-AI et al., 2024) using the GSM8K benchmark under the non-thinking 0-shot evaluation protocol (except for the vanilla model, which uses its native reasoning mode). This helps to further demonstrate the scalability and broad applicability of LoRA-FA.

Table 12: Performance on larger base models (GSM8K, non-thinking 0-shot).

| Method | GSM8K Accuracy | |
|---|---|---|
| | DeepSeek-v2-lite-chat | Qwen3-8B |
| Vanilla (Reasoning mode) | 58.1 | 76.9 |
| LoRA (Non-Thinking 0-shot) | 66.5 | 82.4 |
| LoRA-Pro (Non-Thinking 0-shot) | 71.2 | 84.5 |
| LoRA-FA (Non-Thinking 0-shot) | 71.1 | 84.5 |

These additional experiments on Qwen3-8B and DeepSeek-V2-Lite-Chat (16B MoE) in Table 12 provide robust evidence that LoRA-FA scales effectively to modern, large-scale models and consistently delivers state-of-the-art performance. Notably, LoRA-FA matches the performance of LoRA-Pro on both models, demonstrating that our proposed method is competitive with the latest PEFT approaches, even at scale.

### E.5 Combination with Memory Optimizations

LoRA-FA can be naturally combined with advanced memory optimization approaches, including weight quantization like QLoRA (Dettmers et al., 2023), weight sharding like ZeRO (Rajbhandari et al., 2020), and selective activation recomputation like FlashAttention (Dao et al., 2022).

**Weight quantization.** As discussed before, the memory cost for model weight in 16-bit format is $2n$, where $n$ is the number of model parameters. For example, the model weight memory cost is 130GB for a LLaMA-65B model, which cannot be held in one NVIDIA A100 (80GB) GPU. In LoRA-FA, as the model weights are frozen during fine-tuning, we can quantize them into lower bit width to reduce the model weight memory overhead without affecting the fine-tuning performance. For example, 8-bit (Dettmers et al., 2022) and 4-bit quantization methods (Dettmers et al., 2023) can be combined with LoRA-FA to reduce the model weight memory by 2 and even 4 times.

**Weight sharding.** When training a LLM on multiple GPUs with data parallelism, weight sharding or ZeRO stage-3 (Rajbhandari et al., 2020) technique can be combined with LoRA-FA to shard the model weight into different GPUs, so that the per-GPU memory cost is reduced by the number of GPUs. Different from using ZeRO stage-3 in full-parameter fine-tuning, we only shard the model weights and all-gather them to support the feed-forward and back-propagation computations, without sharding the adaptor related weights and their gradients and optimizer states. However, weight sharding has introduced expensive weight gathering communication cost in LoRA-FA, while data parallelism only communicates a small amount of gradients for trainable parameters.

**Selective activation recomputation.** The activation memory overhead exists in other components of a transformer model, such as attention, layernorm, GeLU, and dropout (Korthikanti et al., 2023). To address it, we can use full activation recomputation to store the input of each transformer block. However, it will disable the memory advantage of LoRA-FA over LoRA, as there is no need to store the inputs of LoRA layers with full activation recomputation. To balance the activation cost and recomputation cost, we instead use selective activation recomputation to recompute only a fraction of model components. For example, FlashAttention (Dao et al., 2022) can eliminate the memory cost of attention softmax outputs and accelerate the attention computations with less HBM accesses. Besides, we can recompute the dropout by storing the random generator state to get the exact mask.

### E.6 Limitations

LoRA-FA also has some limitations: (i) LoRA-FA can only eliminate the activation associated with matrix $A$. While this reduction reaches the theoretical lower bound for removable activations associate with trainable module, however each Transformer layer will still produce its own activations (e.g., the inputs and outputs of attention layers), which LoRA-FA currently cannot reduce. (ii) LoRA-FA remains a low-rank fine-tuning method. As such, when the base model has limited capacity or when the dataset imposes high demands on model performance, it may lead to suboptimal results.

## Impact Statement

This paper presents research aimed at advancing the field of machine learning. While the work may have various potential societal implications, we do not identify any that require explicit emphasis at this time.

