# OpenReview forum: "LoRA-FA: Efficient and Effective Low Rank Representation Fine-tuning"
_ICLR.cc/2026/Conference — Submitted to ICLR 2026_

### Official Review · Reviewer_VK8C · 2025-10-15

**Soundness:** 2
**Presentation:** 3
**Contribution:** 3
**Rating:** 6
**Confidence:** 4

**Summary:**

The paper presents an improvement over LoRA, in which only the B matrix is trained, while the A matrix is frozen, and also a fix is applied to the gradients of layer B to mimic the gradients as if both AB layers are trained. Freezing the A matrix eliminates the need to store the input activations to layer A, thereby reducing memory footprint.

**Strengths:**

* A decent number of LoRA variations were compared
* Reduction in activation memory footprint compared to LoRA
* Method is simple and clear
* Theoretical claims are interesting
* Gradient fix is novel and interesting

**Weaknesses:**

* In the abstract and in the body authors say: “However, LoRA's performance often lags behind Full-FT due to limitations of the low-rank gradient”. This is not always true. In fact, see a quote from the original LoRA paper: “As shown in Table 4, LoRA matches or exceeds the fine-tuning baseline on all three datasets.” .


* Line 52: “However, these studies generally overlook the fact that A and B contribute
asymmetrically”. Not always true, for example see the PRILoRA paper, which prunes only the A matrix, due to the same reason, that the A matrix ‘matters less’. You may want to address this paper.

* Figure 1. (a) in Full FT maybe the drawing of matrices A,B should be removed? Unless you define full FT as such that includes them frozen.

* THM 3.1 says that A*B* \approx A_0 B’*, but the proof doesn’t prove that. The proof shows that in the general case (when they don’t span the same subspace) the Frobenius norm of the difference is r(m-r), which is not zero, nor necessarily small. I may have missed something?

* Line 188, maybe the authors meant dL?

* Table 2 and the other tables: better to have the same amount of decimal places even for round numbers, to keep numbers aligned: 0. -> 0.0

* Table 2: Results are very different from the ones in the original LoRA paper. E.g. In the RTE column (and others) FT got  77.1, LoRA 74.8 (meaning LoRA inferior), while in the LoRA paper FT got 78.7 and LoRA 86.6. (LoRA superior). This is quite a difference. I assume it is because in the LoRA paper they had unique hyperparams for each benchmark, while the authors set a fixed set of hyperparameters for all benchmarks, which are only optimized for the proposed method, and not for the LoRA baseline. Not sure it is a fair comparison. Two things authors can do: (1) For each benchmark from the GLUE, compare the best LoRA-FA hyper-swept model against best LoRA hyper-swept model (2) Compete against LoRA using LoRA best hyperparameters for each benchmark. If authors succeed in at least one of the two, that would be far more convincing, as these benchmarks are very sensitive to hyperparameters.

* Even on authors table 3, on MT-Bench, LoRA gets 5.6 while full-FT gets 5.3, so how can authors claim that full-FT is always better that LoRA?

* In all tables, a column with a number of trainable parameters should be added. Table 3 has rows with different low ranks, which does not make fair comparison.

* Line 668 says A*=A_0 C, so why (9) has \approx  and full equality?

* Line 700, I understand that A_0 is N(0,1), but why A*? If this assumption is for the sake of ease of analysis, maybe it should be addressed. In line 753 authors proved the expectancy of the Frobenius norm, but the theorem said nothing about the Frobenium norm… So the reader is left in suspense, seeking to understand what the answer means.


* A very interesting ablation test is missing: What would happen if A is frozen, but without the gradient fix. I am interested to know how crucial it is.

**Questions:**

See the weaknesses section

---

> ### Author Response · Authors · 2025-11-22
> **Rebuttal**
>
> First, we thank the reviewer for the thoughtful and detailed feedback. Your observations are insightful, and we have revised both the writing and experiments to address them in the updated version.
>
> >Regarding LoRA’s Relative Performance and Asymmetric LoRA Works (W1, W2, W8)
>
> We agree that LoRA can outperform full fine‑tuning on smaller tasks, as confirmed by the original LoRA paper. Our experience and other community reports also show this pattern. However, for more complex tasks including mathematical reasoning, coding, that LoRA often underperforms Full‑FT. That said, the performance gap narrows as the base model’s intrinsic capability increases.
>
> Therefore, we have rephrased the statement in the paper: rather than claiming LoRA is “always worse,” we now emphasize that LoRA is typically competitive on simple tasks but may fall behind Full‑FT on complex reasoning tasks.
>
> As for the asymmetry remark, our intention was to note that among gradient approximation based LoRA variants, few have actually leveraged asymmetry in the optimization structure itself, which distinguishes LoRA‑FA.
>
> >Paper Writing, Organization, and Table Consistency (W3, W5, W6, W9, W10)
>
> We have carefully revised the paper to improve clarity and consistency (including consistent decimal alignment across tables).
>
> Regarding trainable parameters, we will explicitly include this metric in the final revision. Note that because only B is trainable in LoRA‑FA, the total trainable parameters are half that of symmetric LoRA at the same rank r.
>
> >Clarification of Theorem 3.1 and Related Proofs (W4, W11)
>
> We agree that the original statement of Theorem 3.1 might have implied that the approximation is numerically tight. Our intention, however, is to emphasize a structural equivalence rather than an exact or vanishing approximation error. Specifically, the result shows that the LoRA update can be reparameterized through a linear transformation.
>
> To clarify this interpretation, we have revised the theorem and added Remark 1 after its proof. The new remark explicitly explains that the approximation is structural, not numerical, and discusses the expected projection residual. As further explained in the remark, when $r \ll m$ (as is typical in LoRA), the relative per-dimension discrepancy is small, and $A_0 C^\*$ retains most of the subspace structure of $A^*$. Therefore, the collapsible single-layer structure remains valid in practice.
>
> We agree that the Gaussian assumption for $A^\*$ should be explicitly clarified. In the theoretical analysis (Part II of the proof), we assume both $A_0$ and $A^\*$ follow N(0, 1) mainly for analytical tractability and to capture the typical-case behavior of randomly initialized matrices. As shown in Figure 3, the empirical distribution of $A^\*$ indeed follows a Gaussian pattern, supporting this assumption. Please refer to Remark 2 at the end of the proof for a corresponding clarification.
>
> >Ablation: LoRA‑FA Without Gradient Approximation (W12)
>
> Freezing A without applying the gradient approximation mechanism reduces LoRA‑FA to a simple “train‑B‑only” variant. This corresponds to the early 2023 version of LoRA‑FA released on arXiv, which contained no gradient correction step.
>
> We conducted this ablation, and the summarized results (on GSM8K using Llama‑3‑8B) are:
>
> | Method                             | Llama3-8B GSM8K |
> | ---------------------------------- | --------------- |
> | LoRA                               | 71.3            |
> | LoRA-FA w/ gradient approximation  | 70.9            |
> | LoRA-FA w/o gradient approximation | 75.6            |
>
>
> >Future Work on Benchmark‑Specific Hyperparameters (W7)
>
> Due to time constraints, we did not fully hyperparameter‑tune LoRA across every GLUE subtask.
> All GLUE results were obtained using a single hyperparameter configuration tuned on MRPC and then applied uniformly. We agree that further task‑specific tuning would strengthen the comparisons, and we plan to conduct this extended hyperparameter sweep in our future revision.

---

> > ### Comment · Reviewer_VK8C · 2025-11-23
> >
> > Thank you for taking the time to provide your responses; I appreciate the authors’ effort in addressing the questions.

---

### Official Review · Reviewer_qZh5 · 2025-10-25

**Soundness:** 2
**Presentation:** 2
**Contribution:** 2
**Rating:** 2
**Confidence:** 5

**Summary:**

This paper proposes a parameter-efficient fine-tuning algorithm called LoRA-FA, which freezes the A matrix in LoRA and only trains matrix B. To better mimic LoRA's update, the authors provide the optimal closed-form estimation of B's update direction via Theorem 4.1. The authors also conduct experiments to validate LoRA-FA's performance, where LoRA-FA can outperform vanilla LoRA and several variants.

**Strengths:**

1. I believe the major strengths of this work should be establishing the theoretical relationship between LoRA and algorithms like LoRA-FA which only trains one LoRA component at a time, which corresponds to Theorems 3.1 and 4.1.
2. The proposed algorithm has shown noticeable advantage compared to vanilla LoRA in the experiments.

**Weaknesses:**

1. I may consider the core mechanism of LoRA-FA, i.e., only training one LoRA component at a time, not novel because I have already seen a lot of algorithms that use this structure. For example, Flora [arXiv:2402.03293] and LoQT [arXiv:2405.16528] directly apply this structure; GaLore [arXiv:2403.03507] projects the gradient into subspaces and is later shown by GoLore [arXiv:2410.11289] that they are essentially equivalent to this structure. However, none of these related works are discussed or compared in this paper, and I may consider that the authors are not aware of them.
2. Based on my first point, I believe this paper needs stronger theoretical insights or empirical performance to be accepted. Specifically,
i) In theory, the authors only study why LoRA-FA can mimic vanilla LoRA's update via Theorems 3.1 and 4.1. These results, while having their own values, cannot reflect why LoRA-FA can have better performance than vanilla LoRA in practice. As shown in Tables 2 and 3, vanilla LoRA is far worse than full fine-tuning and other variants. Consequently, LoRA-FA's ability to mimic LoRA cannot demonstrate why it can be comparable to stronger baselines.
ii) In practice, stronger baselines, including ReLoRA [arXiv:2307.05695] and other similar approaches that I have mentioned in my first point, are not compared with. This raises a fundamental question: whether LoRA-FA can truly beat previous SoTA algorithms, where some of them even share similar structures?

**Questions:**

I find the algorithmic description in Appendix B not convenient to read. I suggest replacing long variable names with simpler math symbols, and this description should also be included in the main text rather than in the appendix.

---

> ### Author Response · Authors · 2025-11-22
> **Rebuttal**
>
> >Algorithmic description in Appendix B
>
> We have revised and simplified the algorithmic description in the updated version. Variable names are now concise mathematical symbols.
>
> >On Asymmetric LoRA-Related Works
>
> We thank the reviewer for drawing attention to these recent developments. However, we would like to clarify that LoRA-FA was first released on arXiv in August 2023, several months before all the cited works. To the best of our knowledge, LoRA-FA is the first work to propose an asymmetric LoRA fine-tuning method, which leverages asymmetry in the LoRA structure without compromising accuracy.
>
> Since the release of LoRA-FA, many subsequent studies have adopted training asymmetries or subspace projection strategies that can be regarded as special cases or extensions of our formulation. The continued emergence of such asymmetric methods (even into early 2026) underscores the novelty and influence of the original LoRA-FA concept.
>
> Timeline of related works:
>
> ```
> 2023-08 LoRA-FA (ours)
> 2024-02 FLoRA
> 2024-03 GaLore
> 2024-05 LoQT
> 2024-10 GoLore
> ```
>
> It should also be noted that GoLore is not strictly an asymmetric LoRA method; it alternates updates and samples gradients from a uniform distribution on the Stiefel manifold subspace, differing conceptually from LoRA‑FA’s structural asymmetry.
>
> We appreciate this feedback and will explicitly cite and discuss these papers in the final revision, emphasizing the conceptual relationships and differences.
>
> >Why LoRA‑FA Can Be Comparable to Stronger Baselines
>
> The theoretical justification is twofold:
> 1. Theorem 3.1 demonstrates that LoRA‑FA possesses convergence behavior comparable to that of standard LoRA by proving their approximation equivalence.
> 2. Theorem 4.1 extends this by providing a gradient approximation algorithm that targets the full gradient (not a low‑rank projection).
>
> In short, for the performance, $\text{Theorem 3.1: LoRA-FA} \approx \text{LoRA}$, $\text{Theorem 4.1: LoRA-FA} > \text{LoRA}$.
>
> Thus, LoRA‑FA not only matches LoRA in convergence theory but also introduces an enhanced gradient approximation mechanism that contributes to improved empirical performance.
>
> >Comparison with Other Baselines (Including ReLoRA)
>
> We acknowledge the reviewer’s point that ReLoRA is a valuable baseline. However, ReLoRA primarily focuses on iterative re‑initialization and does not belong to the same gradient approximation category as LoRA‑FA. Hence, we prioritized comparisons with stronger baselines that share a more direct conceptual relation to our approach, that those involving asymmetric parameter updates or low‑rank subspace projections.
>
> >Final Note
>
> We highly value your opinion, and we would like to understand more about the key considerations that led to the current rating (2) and whether there might be an opportunity to reconsider the rating after reviewing our clarifications. Your feedback will help us improve both this work and our future research.
>
> Since the initial release of LoRA‑FA in 2023, the method has undergone substantial development and extensive empirical validation demonstrating its effectiveness and efficiency. It has also been successfully integrated into the PEFT library, reflecting its practical adoption and relevance to the community.
>
> We hope that the detailed rebuttal above has addressed your concerns and clarified the contributions and strengths of our work.

---

> > ### Comment · Reviewer_qZh5 · 2025-11-23
> > **Thanks for your response.**
> >
> > I sincerely thank the authors for the detailed and patient response, which has really solved some of my biggest concerns.
> >
> > Specifically:
> >
> > 1. Regarding novelty. Although this paper is claimed to have been posted to arXiv two years ago, I must admit that I have not and could not search for it on arXiv as a reviewer. If true, I believe it is acceptable to claim this work as the first to propose such a framework, while including recent progress on this framework.
> > 2. Regarding the theory. I now understand the theory. Thanks for the explanation, and sorry for my previous misunderstandings.
> >
> > In general, I would like to raise my score in response to the resolved concerns. However, I may not give a rating like 6 because this method may somehow be outdated and not state-of-the-art at present, if compared with recent advancements in this field.

---

### Official Review · Reviewer_UxZa · 2025-11-01

**Soundness:** 2
**Presentation:** 3
**Contribution:** 2
**Rating:** 2
**Confidence:** 4

**Summary:**

The paper introduces LoRA-FA (LoRA by Fixing A), a parameter-efficient fine-tuning method for large language models (LLMs). Standard LoRA adapts model weights using two trainable low-rank matrices, A and B. However, this doubles the number of adaptation parameters and can cause optimization instability. LoRA-FA addresses this by fixing A as a non-trainable matrix (initialized randomly or orthogonally) and only learning B, which effectively reduces trainable parameters by half while maintaining or improving model performance.

**Strengths:**

- The paper is clearly written and easy to follow, with only a few minor typos.
- The proposed method, LoRA-FA, is conceptually simple and straightforward to implement.
- Experimental results demonstrate that LoRA-FA achieves comparable or slightly better performance than standard LoRA across multiple LLMs (e.g., LLaMA, RoBERTa) and benchmarks.

**Weaknesses:**

- Limited Related Work Discussion: The related work section focuses mainly on low-rank adaptation methods (e.g., LoRA) and omits many state-of-the-art PEFT approaches from the past two years. The discussion should be expanded to include subset-of-parameter and sparse fine-tuning methods (e.g., [1–8]). These techniques represent a major trend in recent PEFT research and would help strengthen the paper’s contextual foundation.
* Incremental Novelty: The core idea—freezing one of the LoRA matrices (A)—is relatively incremental regards to novelty.
* Marginal Gains: Reported improvements in performance, memory efficiency, and training speed compared to standard LoRA appear modest. The empirical advantages may not be sufficient to justify the new method as a significant advancement.


### Minor Typos
* “fine-tuning pre-trained LLMs has been shown very effective” → missing “to be”
* “Experiments on system efficiency shows” → should be “show”
* “fix to 64” → should be “fixed to 64”
* “while keep the low-rank matrix A frozen” → should be “while keeping …”
* Several table cells missing digits (e.g., Table 2: “83.3 ± 0.”, “90.1 ± 0”). If intended as ±0, remove the decimal point.
* Caption: “Please ref to Appendix B” → should be “Please refer to Appendix B.”


[1] Parameter-Efficient Fine-Tuning without Introducing New Latency

[2] Sparse Matrix in Large Language Model Fine-tuning

[3] The Lottery Ticket Hypothesis: Finding Sparse, Trainable Neural Networks

[4] Parameter-Efficient Transfer Learning with Diff Pruning

[5] Training Neural Networks with Fixed Sparse Masks

[6] Diff prunning: Parameter-Efficient Transfer Learning with Diff Pruning

[7] Scaling Sparse Fine-Tuning to Large Language Models

[8] Composable Sparse Fine-Tuning for Cross-Lingual Transfer

**Questions:**

- Consider adding evaluations on more recent LLMs beyond the RoBERTa and LLaMA families to strengthen the empirical evidence.
- Include at least one sparse or subset-based PEFT baseline for comparison.
- On Line 408: “LoRA-FA reduces memory usage by more than 27.8 GB (80 GB vs. 52.2 GB)”. On my understanding, when applying LoRA to LLaMA model, the activation memory is trivial since the number of trainable parameters are much less than full fine-tune. I conduct a simple calculation below and the activation memory cost is much smaller than 80GB. Could the author explain more details about how they calculate the activation memory and get the numbers?

Using the LoRA activation memory formula:

**Activation Memory = 2 × b × s × (d + r)**

where:
- **b** = batch size = 16
- **s** = sequence length = 512
- **d** = hidden size = 4096
- **r** = LoRA rank = 64

Substituting the values:

**2 × 16 × 512 × (4096 + 64) = 68,157,440 elements ≈ 130 MB per layer (FP16).**

Since **LLaMA-7B** has **32 transformer layers**, the total activation memory is:

**130 MB × 32 = 4.16 GB.**

Therefore, **LoRA (rank = 64, seq_len = 512, batch = 16)** requires approximately **4.2 GB** of activation memory in total.

---

> ### Author Response · Authors · 2025-11-22
> **Rebuttal**
>
> >Paper writing improvement
>
> We have corrected typos in the revised version.
>
> >Evaluations on recent models
>
> As reported in Appendix E.4, we have included GSM8K evaluations on two more recent models: DeepSeek-v2-lite-chat and Qwen3-8B. The results are restated below:
>
> | Method                         | DeepSeek-v2-lite-chat | Qwen3-8B |
> | ------------------------------ | --------------------- | -------- |
> | Vanilla (Reasoning mode)       | 58.1                  | 76.9     |
> | LoRA (Non-Thinking 0-shot)     | 66.5                  | 82.4     |
> | LoRA-Pro (Non-Thinking 0-shot) | 71.2                  | 84.5     |
> | LoRA-FA (Non-Thinking 0-shot)  | 71.1                  | 84.5     |
>
> >Comparison with Sparse Fine-Tuning Baseline (LoSA)
>
> We chose LoSA (ICLR 2025) as a recent sparse LoRA fine-tuning baseline for comparison with LoRA-FA. LoSA jointly prunes both the base weights W and the low-rank modules, ensuring the final merged model maintains global sparsity. The GSM8K results on different models are shown below:
>
> | Method                                   | Llama2-7B | Llama3-8B | DeepSeek-v2-lite-chat |
> | ---------------------------------------- | --------- | --------- | --------------------- |
> | LoSA w/ 50% sparsity (Huang et al. 2025) | 34.2      | 68.3      | 56.9                  |
> | LoRA-FA (Ours)                           | 57.3      | 75.6      | 71.1                  |
>
> >Clarification on Activation Reduction (Part I)
>
> First, we clarify that when applying LoRA to LLama, activation memory does not decrease, in fact, it typically increases slightly due to the structure of LoRA itself.
>
> Since LoRA only modifies the base weight W, we analyze activation memory in terms of W and the LoRA modules:
>
> 1. In full fine-tuning, the original weight W requires activations.
> 2. When LoRA modules are attached, W is frozen, but A in the LoRA module requires the same activation (input X) as W, while B also requires activation.
> 3. Therefore: LoRA activation = A activation + B activation > full fine-tuning activation (W activation).
>
> The number of trainable parameters affects only the gradients and optimizer states, not this activation analysis.
>
> In practice, other components (e.g., Transformer layers and other modules) contribute activations as well, but our discussion focuses specifically on W and LoRA modules. Details on memory usage for individual linear layers and full fine-tuning are reported in Table 1, Appendix C.1, and Appendix C.2 of the paper.
>
> >Clarification on Activation Reduction (Part II)
>
> In this section, we provide a detailed calculation of how LoRA-FA reduces activation memory compared with LoRA (see Appendix C.3).
>
> In Llama2-7B, the Attention module consists of four linear layers: Query, Key, Value, and Output, each of dimension $d \times d$. The MLP module has three linear layers of dimensions $d \times 3d$, $3d \times d$, and $d \times d$. Consequently, LoRA-FA reduces the activation memory by a total of $18bsd$ per layer. When the number of layers is $L$, LoRA-FA reduces activation memory by at least $18bsdL$ bytes compared to LoRA.
>
> For instance, when using a batch size of 8 and a sequence length of 1024, substituting $d = 4096$ and $L = 32$ for Llama2-7B, the theoretical savings of LoRA-FA over LoRA amount to at least 18 GB (18 * 8 * 1024 * 4096 * 32 /1024/1024/1024) of activation memory. In practice, due to PyTorch's memory reservation behavior or the retention of activations by other functions, the actual memory reduction achieved by LoRA-FA often exceeds this theoretical estimate.
>
> >Final Note
>
> We highly value your opinion, and we would like to understand more about the key considerations that led to the current rating (2) and whether there might be an opportunity to reconsider the rating after reviewing our clarifications. Your feedback will help us improve both this work and our future research.
>
> Since the initial release of LoRA‑FA in 2023, the method has undergone substantial development and extensive empirical validation demonstrating its effectiveness and efficiency. It has also been successfully integrated into the PEFT library, reflecting its practical adoption and relevance to the community.
>
> We hope that the detailed rebuttal above has addressed your concerns and clarified the contributions and strengths of our work.

---

### Official Review · Reviewer_JPBd · 2025-11-10

**Soundness:** 3
**Presentation:** 2
**Contribution:** 2
**Rating:** 4
**Confidence:** 4

**Summary:**

To improve the efficiency and effectiveness of parameter-efficient fine-tuning (PEFT) for large language models (LLMs), the authors propose LoRA-FA, a simplified yet expressive variant of LoRA. By revealing an asymmetric, collapsible structure in LoRA updates, the authors show that one projection matrix can be frozen without losing expressivity. LoRA-FA freezes the projection-down matrix and trains only the projection-up matrix, further introducing closed-form gradient corrections to better approximate full gradients. Experiments on benchmarks such as GLUE, GSM8K, and HumanEval demonstrate that LoRA-FA matches or exceeds the performance of existing PEFT and Full-FT, while significantly reducing activation memory and computational overhead.

**Strengths:**

* S1: This paper provides a new perspective on LoRA optimization by interpreting its update as a collapsible single-layer linear regression and proposing a one-sided training method (training only matrix B).
* S2: The paper is well-structured and clearly presented, with theoretical proofs supporting the proposed method.
* S3: Extensive experiments are conducted across various tasks—including NLU (GLUE), math, code, and dialogue—comparing against several recent PEFT/LoRA variants (e.g., LoRA-GA, LoRA-Pro, AdaLoRA), demonstrating the method’s effectiveness and efficiency.

**Weaknesses:**

* W1: The assumptions in Theorem 3.1 / 4.1 may be too strong, and the paper lacks a discussion of the failure boundaries under practical model settings (e.g., when the rank $r$ is small or $A$ is nearly singular), as well as the consequences when these assumptions are not satisfied.
* W2: There is no comprehensive quantitative analysis of the additional computational or communication overhead introduced by the proposed method.
* W3: The paper lacks sensitivity analyses for some key hyperparameters, such as the scaling factor $\alpha$ and the effect of freezing different proportions of $A$ (e.g., only freezing certain layers instead of “all linear layers”).

**Questions:**

* Q1: Could the authors further elaborate on whether these assumptions still hold in practical models (e.g., when the rank is very small or $A$ is nearly singular)? How would the closed-form solution and performance of LoRA-FA be affected when $A$ is rank-deficient or poorly conditioned?
* Q2: The paper highlights that LoRA-FA can reduce activation memory, but it does not provide detailed data on the additional computational and communication costs. Could the authors include such quantitative analysis?
* Q3: Have the authors tried applying LoRA-FA only to specific layers (such as attention or MLP layers), or mixing LoRA-FA with standard LoRA across different layers?

---

> ### Author Response · Authors · 2025-11-22
> **Rebuttal**
>
> >Impact of A and Rank on Convergence (Q1 / W1)
>
> For relatively simple tasks (e.g., datasets such as GLUE), setting the rank to 8 or 16 is generally sufficient. For more complex tasks (such as those in the mathematical domain), a rank of 64 or 128 is typically appropriate. Users need only select from these two sets of hyperparameters based on task complexity, without the need to exhaustively search over other rank values.
>
> Moreover, since LoRA-FA exhibits limited sensitivity to the rank in terms of memory consumption, it is always feasible to use higher LoRA ranks, which may potentially yield improved performance. In Table 3 of our paper, we specifically compare the performance at ranks 64 and 128, and the results demonstrate that performance remains consistently strong at rank 128. Therefore, for users employing LoRA-FA, we recommend directly adopting a rank of 128, which is also the practice we currently follow.
>
> Regarding matrix conditioning: since the LoRA‑FA matrices A are initialized using Kaiming uniform or normal distributions, pathological or ill‑conditioned A values are largely avoided under standard initialization schemes. In other words, the theoretical assumptions made in Theorems 3.1 and 4.1 remain stable in practice with reasonable initialization.
>
> >Computational and Communication Overhead (Q2 / W2)
>
> As illustrated in Figure 2(b), LoRA‑FA introduces no significant computational overhead compared with standard LoRA. The reasoning is as follows:
>
> 1. Pre‑computation Phase. The operations involving A (e.g., transposes and inverses) are computed once before training, not during each iteration. This ensures the runtime efficiency of LoRA‑FA matches that of LoRA.
> 2. Training Phase. During training, LoRA‑FA uses the pre‑computed matrices to adjust the gradients of trainable weights B. While this introduces a minor additional step, LoRA‑FA simultaneously avoids computing the gradient of A. These two effects cancel out, resulting in throughput that is on par with standard LoRA.
> 3. Communication Cost. LoRA‑FA does not introduce extra communication overhead. The gradient adjustment depends on matrix A, which is identical across devices (e.g., in data parallelism), meaning there is no need to communicate transposed factors. Under model parallel (MP) or tensor parallel (TP) setups, LoRA modules are generally small enough that both W and AB can be placed on the same device, thus avoiding additional inter‑device communication.
> 4. Memory Impact of Pre‑computation. The pre‑computed results must be stored, but the overhead is negligible. For example, in Llama3‑8B with rank = 128, the estimated memory for storing pre‑computed matrices is $(128\times128\times4+128\times128\times3)\times32\times2/1024/1024=7\text{MB}$, which is practically insignificant.
>
> >Applying LoRA‑FA to Specific Layers (Q3 / W3)
>
> We conducted preliminary experiments to evaluate partial attachment of LoRA‑FA across model layers. The findings are summarized below:
>
> | Method                  | Llama3-8B GSM8K |
> | ----------------------- | --------------- |
> | LoRA-FA all layers      | 75.6            |
> | LoRA-FA MLP & LoRA Attn | 73.5            |
> | LoRA all layers         | 71.3            |
>
> We observed that LoRA‑FA achieves its best performance when applied to both attention and MLP layers simultaneously. The mixed configuration (LoRA‑FA in MLP, standard LoRA in attention) results in a mild but consistent performance drop. Although suboptimal, this outcome is interesting and suggests possible directions for hybrid fine‑tuning in future work.

---

### Meta-Review · Area_Chair_aXpT · 2026-01-07

**Summary:**

While the authors addressed several technical questions and added clarifications, two key concerns remain insufficiently resolved.

**Novelty and Positioning:** The novelty and positioning of LoRA-FA relative to existing asymmetric or subspace-based LoRA variants remain unclear. Although the rebuttal emphasizes the early release timeline and structural asymmetry of LoRA-FA, reviewers (specifically Reviewer UxZa and Reviewer VK8C) still note substantial conceptual overlap with prior and concurrent methods that train only one LoRA factor or project gradients into low-dimensional subspaces (e.g., PRILoRA, LoSA). The rebuttal does not fully establish why LoRA-FA should be viewed as a distinct methodological advance rather than part of an emerging family of similar approaches.

**Empirical Fairness and Robustness:** Despite added experiments and explanations, reviewers remain unconvinced that the reported gains are consistently robust across tasks and hyperparameter settings. In particular, the use of shared or unevenly tuned hyperparameters (Reviewer JPBd), sensitivity to rank and gradient correction choices, and limited comparisons with stronger or more recent baselines (Reviewer UxZa) weaken the strength of the empirical claims.

Overall, these unresolved issues regarding novelty and experimental robustness limit the paper’s impact.

**Reviewer Concerns:**

While the authors resolved several technical points, added comparison baselines, and softened the claims regarding LoRA always lagging behind Full-FT, the key issues regarding novelty, hyperparameter robustness, and theoretical rigor remained. In particular, Reviewer VK8C and Reviewer JPBd pointed out that the assumptions (e.g., rank-deficient $A$, Gaussian distributions) might not hold in all practical settings, and the "structural equivalence" argument may not fully justify the approximation errors in the gradient correction.

**Reviewer Scores:**

Reviewer UxZa explicitly noted the willingness to raise the score (likely from 2 to 4), but other reviewers likely would have maintained their scores.

---

### Decision · Program_Chairs · 2026-01-26

Reject